# OmniCast: A Masked Latent Diffusion Model for Weather Forecasting Across Time Scales

**Tung Nguyen[1]   Tuan Pham[2]   Troy Arcomano[3,4]   Veerabhadra Kotamarthi[3]**
**Ian Foster[3]   Sandeep Madireddy[3]   Aditya Grover[1]**
[1]UCLA   [2]UCI   [3]Argonne National Laboratory   [4]Allen Institute for AI

## Abstract

Accurate weather forecasting across time scales is critical for anticipating and mitigating the impacts of climate change. Recent data-driven methods based on deep learning have achieved significant success in the medium range, but struggle at longer subseasonal-to-seasonal (S2S) horizons due to error accumulation in their autoregressive approach. In this work, we propose OmniCast, a scalable and skillful probabilistic model that unifies weather forecasting across timescales. OmniCast consists of two components, a VAE model that encodes raw weather data into a continuous, lower-dimensional latent space, and a diffusion-based transformer model that generates a sequence of future latent tokens given the initial conditioning tokens. During training, we mask random future tokens and train the transformer to estimate their distribution given conditioning and visible tokens using a per-token diffusion head. During inference, the transformer generates the full sequence of future tokens by iteratively unmasking random subsets of tokens. This joint sampling across space and time mitigates compounding errors from autoregressive approaches. The low-dimensional latent space enables modeling long sequences of future latent states, allowing the transformer to learn weather dynamics beyond initial conditions. OmniCast performs competitively with leading probabilistic methods at the medium-range timescale while being $10\times$ to $20\times$ faster, and achieves state-of-the-art performance at the subseasonal-to-seasonal scale across accuracy, physics-based, and probabilistic metrics. Furthermore, we demonstrate that OmniCast can generate stable rollouts up to 100 years ahead. Code and model checkpoints are available at https://github.com/tung-nd/omnicast.

## 1   Introduction

Accurate weather forecasting across time scales is essential for anticipating extreme events, managing resources, and mitigating the impacts of climate change. While medium-range forecasting, which encompasses predictions up to approximately two weeks, has seen remarkable progress with both numerical and data-driven approaches, extending prediction skill beyond this horizon remains a significant challenge. Subseasonal-to-seasonal (S2S) forecasting, which aims to predict atmospheric conditions from two to six weeks ahead, represents this next frontier. This timescale bridges the gap between short-term weather forecasts and longer-term climate projections, enabling more informed decision-making for extreme weather events such as droughts, floods, and heatwaves [51, 36, 52, 8]. S2S prediction is particularly challenging due to the interplay between atmospheric initial conditions, essential for short-term and medium-range forecasting, and boundary conditions dominating seasonal and climate predictions [27, 28]. Traditional numerical weather prediction (NWP) models, built upon solving differential equations of fluid dynamics and thermodynamics, have been instrumental in advancing S2S weather prediction [36, 47, 48]. However, numerical methods incur substantial computational costs due to the complexity of integrating large systems of differential equations,

particularly at fine spatial and temporal resolutions. This computational bottleneck also constrains the ensemble size of ensemble systems, which is crucial for achieving accurate S2S predictions.

To overcome the challenges of NWP systems, there has been a growing interest in data-driven approaches based on deep learning for weather forecasting [10, 43, 50]. These approaches involve training deep neural networks on historical datasets, such as ERA5 [14, 15, 39, 40], to learn the underlying weather patterns. Once trained, they can produce forecasts in seconds compared to the hours required by NWP models. Recent deep learning methods such as PanguWeather [2], Graphcast [22], and Stormer [33] have also shown superior accuracy in medium-range weather forecasting, surpassing operational IFS [49], the state-of-the-art NWP system. However, their application to the S2S timescale has been limited [31]. One possible explanation for this limitation is the rapid error compounding in their autoregressive designs, in which a model learns to forecast the future weather state at a small interval and iteratively feeds its prediction back as input to achieve longer-horizon forecasts. Even though previous works have proposed multi-step finetuning to mitigate this issue, back-propagation through a large number of forward passes required for S2S timescales is computationally prohibitive. Moreover, training a neural network to forecast at a small interval only allows the model to learn the initial conditions problem, ignoring boundary conditions that are critical for prediction at S2S timescales.

In this work, we propose OmniCast, a novel latent diffusion model for skillful probabilistic weather forecasting across time scales. OmniCast follows a two-stage training process. In the first stage, we train a VAE model [19] that compresses raw weather data into a continuous, lower-dimensional latent space. In the second stage, we train a transformer to model the distribution of a sequence of future latent tokens given the initial conditioning tokens using a masked generative framework [3, 55]. Specifically, during training, we randomly mask a subset of future tokens, and task the transformer to *unmask* these tokens based on the conditioning tokens and the visible tokens. Since the latent tokens lie in a continuous space, we use a small diffusion network on top of the transformer model to estimate the per-token distribution of unmasked tokens. In addition to the diffusion loss, we apply a mean-squared error (MSE) objective to enforce the model to accurately predict the first few latent frames deterministically. After training, OmniCast generates forecasts for the full sequence of future tokens through an iterative process. In each iteration, the model selects a subset of future tokens to unmask given the conditioning tokens and previously unmasked tokens, continuing this process until all future tokens are generated. The unmasking operation involves sampling from the diffusion model, with the number and positions of tokens selected to unmask in each iteration determined by a predefined schedule and unmasking order. This joint generation of future tokens across time and space significantly mitigates the compounding errors issue of an autoregressive approach. Furthermore, training on the full sequence of future frames enables OmniCast to address both initial condition problems and boundary condition challenges, which are critical for S2S prediction.

We evaluate OmniCast on ChaosBench [31], a recent benchmark for subseasonal-to-seasonal prediction. OmniCast achieves state-of-the-art performance on key atmospheric variables across various accuracy, physics-based, and probabilistic metrics. Additionally, we carefully study the impact of different design choices, including the auxiliary MSE loss, training sequence lengths, unmasking order, and diffusion sampling temperature, on the forecasting performance of OmniCast.

## 2 Related Work

**Data-driven weather forecasting** Deep learning has become a promising approach in the field of weather forecasting. Recent advancements with powerful architectures have achieved significant successes, providing faster inference and superior forecasting accuracy compared to IFS, the gold-standard numerical weather prediction system. Notable methods include FourCastNet [35], which utilizes an adaptive neural operator architecture; Keisler [17]'s, GraphCast [22], and AIFS [24], which leverage graph neural networks; and a series of transformer-based models such as PanguWeather [2], Stormer [33], and others [32, 6, 4, 7]. Beyond deterministic predictions, the field has increasingly focused on probabilistic forecasting to better account for forecast uncertainty. Common approaches involve integrating existing architectures with generative frameworks, including diffusion models [37, 30], normalizing flows [7], and latent variable models [34]. Others explore ensemble predictions through initial condition perturbations, exemplified by methods like AIFS-CRPS [24] and NeuralGCM [20].

**Data-driven S2S prediction** Recent benchmarks have emerged to evaluate data-driven methods at S2S timescales. While many focus on regional forecasts such as the US [16, 29], ChaosBench [31] offers a comprehensive framework for global S2S prediction, providing extensive numerical baselines and physics-based metrics. A key finding from ChaosBench shows that state-of-the-art deep learning methods struggle to extend to S2S timescales. These methods predominantly rely on autoregressive approaches that generate predictions iteratively at short time intervals, leading to error accumulation with increasing lead times. While multi-step finetuning helps mitigate this issue for medium-range forecasts, it becomes computationally prohibitive for S2S predictions due to the extensive number of required forward passes. Moreover, training models with short time intervals fails to capture boundary conditions essential for long-term weather patterns. While Fuxi-S2S [5] was proposed for S2S prediction, it focuses on forecasting daily averaged statistics, which fundamentally alters the underlying weather dynamics and makes it inapplicable to forecasting at instantaneous time steps.

## 3 Background and Preliminaries

### 3.1 Weather forecasting

The goal of weather forecasting is to forecast future weather conditions $X_T \in \mathbb{R}^{V \times H \times W}$ based on initial conditions $X_0 \in \mathbb{R}^{V \times H \times W}$, where $T$ represents the target lead time, $V$ denotes the number of input and output physical variables (e.g., temperature and geopotential), and $H \times W$ corresponds to the spatial resolution of the data, determined by the density of the global grid. In subseasonal-to-seasonal (S2S) forecasting, we focus on lead times ranging from 2 to 6 weeks. Autoregressive modeling is a dominant paradigm in data-driven weather forecasting, where a model iteratively produces forecasts $X_{\delta t}$ at a short interval $\delta t$ to reach the target lead time $T$. In this work, we propose an alternative approach: training a generative model to estimate the distribution of the entire sequence of future weather states $X_{1:T}$ given initial conditions $X_0$. This approach mitigates error accumulation and enables the model to learn both initial and boundary condition dynamics by considering the complete sequence of weather states.

### 3.2 Masked generative modeling

Masked generative modeling is an efficient and powerful approach for image and video generation in computer vision [3, 55, 25]. In this framework, visual data $X_{1:T} \in \mathbb{R}^{T \times V \times H \times W}$ ($T = 1$ for images) is first embedded by a VAE encoder into a sequence of tokens $\mathbf{x} \in \mathbb{R}^{N \times D}$, where $N$ represents the length of the flattened token sequence. During training, we apply a binary mask to randomly select a subset of tokens to be predicted, creating a corrupted sequence. We then train a transformer model to recover the original tokens at masked positions based on both the visible tokens and any additional conditioning information such as initial frames. For generation, the framework employs an iterative decoding process that starts with a fully masked sequence of future tokens. In each iteration, the model predicts a random subset of masked tokens in parallel, where the number and positions of the unmasked tokens follow a predefined schedule and order. This process continues until all tokens are unmasked, at which point the generated tokens are decoded back to the original domain through a VAE decoder. This framework offers key advantages for weather forecasting: it allows the model to capture long-range dependencies across the entire sequence while avoiding the error accumulation typical in autoregressive approaches, and the iterative refinement process enables the model to maintain consistency across both spatial and temporal dimensions.

### 3.3 Modeling continuous tokens with diffusion models

In the masked generative modeling framework, a common practice is to embed the raw visual data into a discrete latent space and train the transformer model using a cross-entropy objective. However, this approach relies on vector-quantized VAE models [45], which are sensitive to gradient approximation strategies [41, 38, 21] and typically achieve lower reconstruction quality than continuous-valued VAEs. Recent works [44, 26] have demonstrated that discretization can be eliminated by directly modeling the per-token probability distribution in a continuous latent space. In this work, we adopt diffusion models for continuous distribution modeling.

Given data $x \in \mathbb{R}^D$ and its conditioning information $z \in \mathbb{R}^D$, we model the conditional distribution $p(x \mid z)$ using a diffusion process that gradually transforms a Gaussian prior into the target distribution.

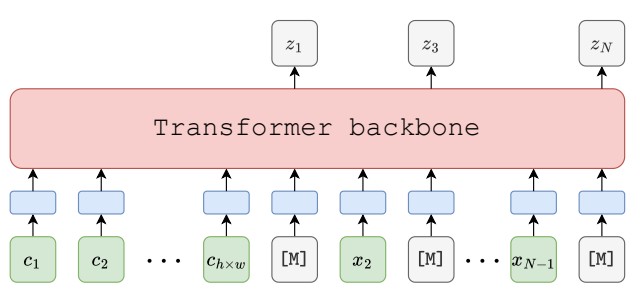

Figure 1: OmniCast processes the latent tokens through a transformer backbone that outputs a vector $z_i$ for each position $i$ in the sequence.

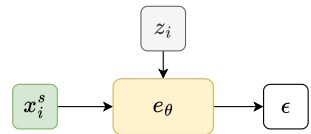

Figure 2: The denoising network $e_\theta$ predicts the noise $\epsilon$ from $z_i$ and $x_i^s$.

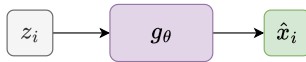

Figure 3: The deterministic network predicts directly $x_i$ from $z_i$.

The forward diffusion process progressively adds Gaussian noise to the data $x$ following:

$$x_s = \sqrt{\alpha_s}x + \sqrt{1 - \alpha_s}\epsilon, \tag{1}$$

where $s$ indicates the diffusion step, $\alpha_s$ determines the noise schedule, and $\epsilon \sim \mathcal{N}(0, \mathbf{I})$ represents Gaussian noise. The reverse process employs a denoising network $\epsilon_\theta(x_s, s, z)$ parameterized by $\theta$ to predict the noise component from the noisy input $x_s$ and condition $z$:

$$\mathcal{L}_{\text{diff}}(\theta) = \mathbb{E}_{\epsilon,x}\left[\|\epsilon_\theta(x_s, s, z) - \epsilon\|^2\right]. \tag{2}$$

At inference time, conditional sampling begins with a random Gaussian noise $x_S \sim \mathcal{N}(0, \mathbf{I})$ and iteratively applies the reverse diffusion process:

$$x_{s-1} = \frac{1}{\sqrt{\alpha_s}}\left(x_s - \frac{1 - \alpha_s}{\sqrt{1 - \bar{\alpha}_s}}\epsilon_\theta(x_s, s, z)\right) + \tau\sigma_s\delta, \tag{3}$$

where $\bar{\alpha}_s = \prod_{k=1}^s \alpha_k$, $\delta \sim \mathcal{N}(0, \mathbf{I})$ and $\sigma_s$ controls the magnitude of noise added at each step. This iterative process generates samples from the learned conditional distribution $p_\theta(x \mid z)$. Following [26], we additionally scale the noise $\sigma_s\delta$ by the temperature $\tau$ that controls the sample diversity from the diffusion model.

## 4 Methodology

We present OmniCast, a novel method for subseasonal-to-seasonal prediction. Similar to previous works in video generation, OmniCast consists of two components: a VAE model that compresses the raw weather data into a lower-dimensional space, and a masked generative transformer model in this latent space. We present the two components and their key design choices in this section.

### 4.1 VAE for weather data embedding

A VAE encoder embeds a weather state $X \in \mathbb{R}^{V \times H \times W}$ into a map of $h \times w$ latent tokens, where $h < H$ and $w < W$. In vector-quantized VAEs, each entry in the latent map is an integer index from a fixed-size vocabulary, representing a discrete latent space. While this discretization is widely adopted in computer vision due to its compatibility with cross-entropy training and straightforward sampling from softmax distributions, it presents significant challenges for weather data. Unlike RGB images with three channels, weather states can contain hundreds of physical variables, resulting in an extreme compression requirement. For instance, consider compressing weather data with 100 variables (32 bits per value) by a factor of 4 in each spatial dimension, using a vocabulary size of $2^{13} = 8192$ (13 bits per latent token). This results in a compression ratio of $(32 \times 100 \times H \times W)/(13 \times (H/4) \times (W/4)) \approx 3938$. Such aggressive compression leads to substantial reconstruction errors, ultimately degrading the performance of the second-stage generative modeling.

Therefore, we adopt a continuous VAE model for OmniCast, where each token in the $h \times w$ latent map is a continuous vector of $D$ dimensions. With $D = 16$, for example, the compression ratio becomes $(32 \times 100 \times H \times W)/(32 \times 16 \times (H/4) \times (W/4)) = 100$, substantially lower than the discrete approach. While it is also possible to compress a sequence of weather states $X_{1:T} \in \mathbb{R}^{T \times V \times H \times W}$ in both temporal and spatial dimensions, our preliminary experiments showed no clear benefits from temporal compression, leading us to adopt per-frame embedding.

## 4.2 Masked generative modeling for S2S prediction

After training the VAE, we embed the initial condition into a sequence of tokens $\mathbf{c} = (c_1, c_2, \ldots, c_{h \times w})$. Similarly, each future weather state is embedded into a sequence of tokens, which are concatenated to form the complete sequence of future tokens $\mathbf{x} = (x_1, x_2, \ldots, x_N)$, where $N = T \times h \times w$ represents the total number of future tokens. Each latent token is a continuous vector of dimension $D$. Our generative modeling objective is to estimate the conditional distribution $p(\mathbf{x} \mid \mathbf{c})$ from the training data.

We achieve this using a masked generative framework, as illustrated in Figure 1. During training, we sample a binary mask $\mathbf{m} = [m_i]_{i=1}^N \sim p_\mathcal{U}$ and replace tokens $x_i$ with a learnable, continuous [MASK] token where $m_i = 1$, creating a corrupted sequence $\overline{\mathbf{x}} = \mathbf{m}(\mathbf{x})$. The generative objective is to estimate the distribution of masked tokens conditioned on the visible and conditioning tokens:

$$\mathcal{L}_{\text{gen}}(\theta) = \mathbb{E}_{\mathbf{m} \sim p_\mathcal{U}} \left[ \sum_{i \text{ s.t. } m_i = 1} - \log p_\theta(x_i \mid \mathbf{c}, \overline{\mathbf{x}}) \right]. \tag{4}$$

The model processes the input by concatenating the conditioning tokens $\mathbf{c}$ with the corrupted future tokens $\overline{\mathbf{x}}$, adding positional encodings to the embedded sequence, and passing it through a bi-directional transformer backbone to obtain vectors $z_i$ for each masked position. Given these vectors, the per-token objective $\log p_\theta(x_i \mid \mathbf{c}, \overline{\mathbf{x}})$ in Equation 4 simplifies to $\log p_\theta(x_i \mid z_i)$. To model this continuous distribution, we employ a diffusion model where $z_i$ serves as conditional information for a denoising network – implemented as a small MLP on top of the transformer (Figure 2). We train the denoising network and the transformer backbone jointly using the diffusion loss specified in Equation 2. Conceptually, this diffusion objective encourages the model to produce representations $z_i$ that facilitate effective denoising.

**Auxiliary deterministic objective** To encourage accurate predictions of near-term future tokens, we incorporate an auxiliary mean-squared error loss in the latent space. We implement this through a separate MLP head that produces deterministic predictions $\hat{x}_i$ from $z_i$, training it jointly with the transformer backbone. Since weather dynamics become increasingly chaotic beyond day 10, making deterministic predictions progressively less meaningful, we apply this loss only to the first 10 future frames. Furthermore, we employ an exponentially decreasing weighting scheme to emphasize the importance of accurate predictions for earlier frames. The deterministic objective is thus:

$$\mathcal{L}_{\text{deter}}(\theta) = \mathbb{E}_{\mathbf{m} \sim p_\mathcal{U}} \left[ \sum_{m_i = 1} w(i) ||x_i - \hat{x}_i||_2^2 \right]. \tag{5}$$

Appendix A.2 presents the details of this objective. The complete training objective combines both losses: $\mathcal{L}(\theta) = \mathcal{L}_{\text{gen}}(\theta) + \mathcal{L}_{\text{deter}}(\theta)$.

**Sampling from OmniCast** At inference time, we generate samples from $p(\mathbf{x} \mid \mathbf{c})$ through an iterative decoding process, starting from a sequence of fully masked future tokens. Each iteration consists of three steps: first, the transformer backbone processes the conditioning tokens and corrupted future tokens to produce vectors $z_i$ for each masked position; second, a subset of masked positions is randomly selected according to a predefined schedule for unmasking; third, for each selected position, the diffusion model generates token $x_i$ by conditioning on $z_i$ and performing a fixed number of diffusion steps. This process iterates until all future tokens are revealed, at which point the VAE decoder maps the generated tokens back to the weather domain. To generate an ensemble of forecasts, we simply replicate the initial tokens and perform independent sampling for each copy. Four hyperparameters affect the sampling procedure: the number of unmasking iterations, the unmasking order, the number of diffusion steps, and the diffusion temperature.

## 4.3 Implementation details

**Architectural details** For the transformer backbone, we adopt the encoder-decoder architecture from Masked Autoencoder (MAE) [13]. The model processes an input sequence in two stages: first, the encoder processes the conditioning and visible tokens; second, the encoded sequence is augmented with learnable [MASK] tokens at appropriate positions and passed through the decoder to produce $z_i$ for each position $i$. Both the encoder and decoder are bidirectional, employing full attention. Before feeding to either the encoder or decoder, we add the input sequences with positional

embeddings that combine two components: temporal embeddings to distinguish different frames, and spatial embeddings to differentiate tokens within each frame. The encoder and decoder follow the Transformer [46] implementation in ViT [9], each having 16 layers with 16 attention heads, a hidden dimension of 1024, and a dropout rate of 0.1.

**Mask sampling** During training, we sample a masking ratio $\gamma \sim \mathcal{U}[0.5, 1.0]$ and generate a corresponding binary mask $\mathbf{m}$, where $\gamma = 0.75$ indicates that $75\%$ of entries in $\mathbf{m}$ are 1. For inference, we start with full masking ($\gamma = 1.0$) and gradually reduce it to 0.0 following a cosine schedule [3]. We set the number of unmasking iterations to match the number of future weather states $T$ by default. We employ random masking orders across both spatial and temporal dimensions for training and inference.

**Diffusion loss details** We use a linear noise schedule with 1000 steps at training time that are resampled to 100 steps at inference. The denoising network $\epsilon_\theta$ is implemented as a small MLP following Li et al. [26]. Specifically, the network consists of six residual blocks, each comprising a LayerNorm (LN), a linear layer, a SiLU activation, and another linear layer, with a residual connection around the block. Each block maintains a width of 2048 channels. The network takes the vector $z_i$ from the transformer as conditioning information, which is combined with the time embedding of the diffusion step $s$ through adaptive layer normalization (AdaLN) in each block's LN layers.

## 5 Experiments

We compare OmniCast with state-of-the-art deep learning and numerical methods on both medium-range and S2S time scales, using WeatherBench2 [40] (WB2) and ChaosBench [31] as benchmarks, respectively, and conduct extensive ablation studies to assess the contribution of each component in OmniCast. We further test the stability of OmniCast up to 100 years ahead in Appendix B.5.

Across both tasks, we train and evaluate OmniCast on 69 variables from the ERA5 reanalysis dataset [15], including four surface-level variables – 2-meter temperature (T2m), 10-meter U and V wind components (U10, V10), and mean sea-level pressure (MSLP), as well as five atmospheric variables – geopotential (Z), temperature (T), U and V wind components, and specific humidity (Q), each at 13 pressure levels {50, 100, 150, 200, 250, 300, 400, 500, 600, 700, 850, 925, 1000} hPa. For medium-range forecasting, we use native $0.25°$ resolution ($721 \times 1440$ grids) and follow WB2 to train on years 1979–2018, validate on 2019, and test on 2020 using initial conditions at 00UTC and 12UTC. For S2S prediction, we downsample the data to $1.40625°$ ($128 \times 256$ grids) and follow ChaosBench to train on 1979–2020, validate on 2021, and test on 2022 using 00UTC initializations.

### 5.1 OmniCast for S2S prediction

**Training and inference details** We train a VAE that embeds each weather state of shape $69 \times 128 \times 256$ into a latent map of shape $1024 \times 8 \times 16$, reducing spatial dimensions by a factor of 16. The architectural details and training process of the VAE are described in Appendix A.1. We train OmniCast to forecast a sequence of $T = 44$ future weather states at 24hr intervals, covering lead times from 1 to 44 days. Each training example consists of $45 \times 8 \times 16 = 5760$ latent tokens, including the initial condition. During inference, we generate the complete future sequence in 44 iterations (1 iteration per frame) using a diffusion temperature of $\tau = 1.3$. We produce an ensemble of 50 forecast sequences for each initial condition.

**Baselines** We compare OmniCast with PanguWeather (PW) [2] and GraphCast (GC) [22], two leading open-sourced deep learning methods, and ensemble systems of four numerical models from different national agencies: UKMO-ENS (UK) [53], NCEP-ENS (US) [42], CMA-ENS (China) [54], and ECMWF-ENS (Europe) [11]. We refer to ChaosBench for details about these baselines. Following ChaosBench, we report results on T850, Z500, and Q700 at lead times from 1 to 44 days. We additionally compare OmniCast with ClimaX [32] and Stormer [33] in Appendix B.2. We do not compare against Fuxi-S2S [5] as Fuxi-S2S forecasts daily average values from past daily averages, making it incomparable with OmniCast and the rest of the methods, which perform point-in-time weather forecasting based on an initial condition. We are also not able to run Gencast [37] and NeuralGCM [20] for S2S due to their significant computational demands.

**Results** Figure 4 compares different methods on three deterministic metrics: Root Mean-Squared Error (RMSE), Absolute Bias (ABS BIAS), and Multi-scale Structural Similarity (SSIM). At shorter

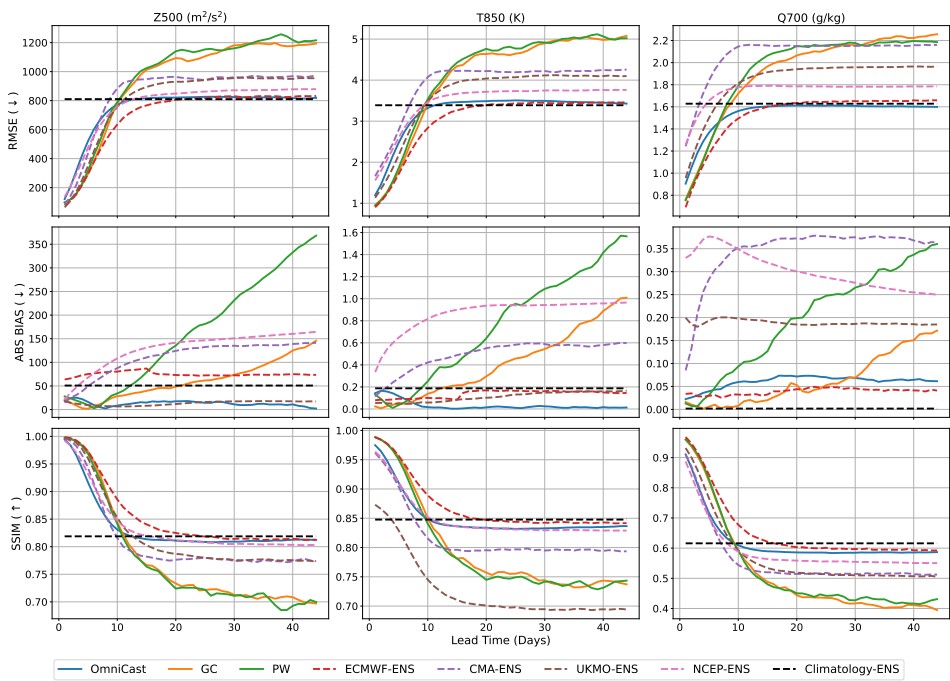

Figure 4: Deterministic performance of different methods at lead times from 1 to 44 days across three key variables. Solid curves are deep learning methods and dashed curves are numerical methods.

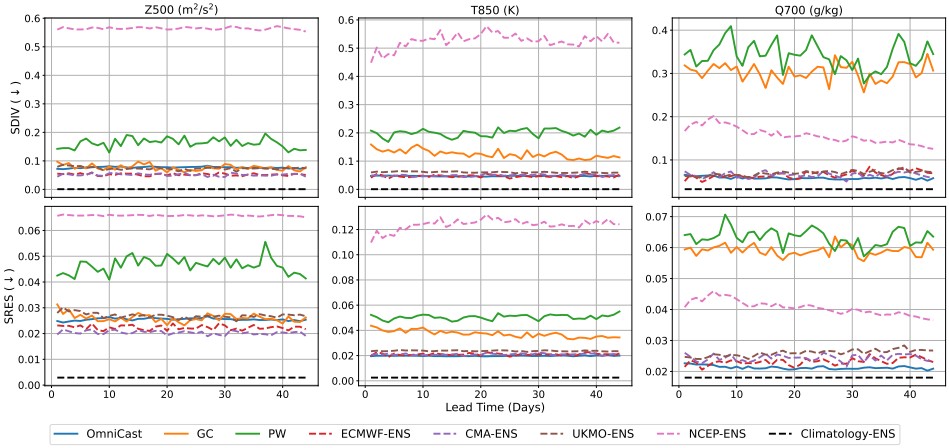

Figure 5: Physics-based metrics of different methods at lead times from 1 to 44 days across three key variables. Solid curves are deep learning methods and dashed curves are numerical methods.

lead times, OmniCast shows slightly worse performance on RMSE and SSIM than other baselines, which is expected since we train OmniCast to model a full sequence of future weather states rather than optimizing for short- and medium-range predictions. However, OmniCast's relative performance improves with increasing lead time, ultimately matching ECMWF-ENS as one of the top two performing methods beyond day 10. Notably, OmniCast demonstrates the lowest bias among all baselines, maintaining near-zero bias across all three target variables.

Physical consistency also plays a crucial role in S2S prediction, particularly for ensemble systems. We evaluate this aspect using two physics-based metrics: Spectral Divergence (SDIV) and Spectral Residual (SRES), which measure how closely the power spectra of predictions match those of ground-truths. As shown in Figure 5, OmniCast achieves substantially better physical consistency than other deep learning methods, and often outperforms all baselines on these metrics. These results demonstrate how OmniCast effectively preserves signals across the frequency spectrum.

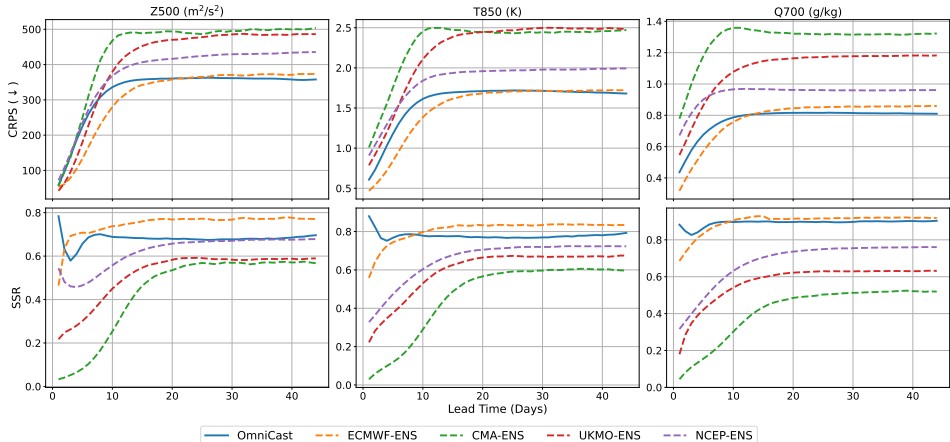

Figure 6: Probabilistic performance of different methods at lead times from 1 to 44 days across three key variables. Solid curves are deep learning methods and dashed curves are numerical methods.

Finally, we compare OmniCast with the four numerical ensemble systems on two probabilistic metrics: Continuous Ranked Probability Score (CRPS) and Spread/Skill Ratio (SSR) (closer to 1 is better). Figure 6 shows that OmniCast and ECMWF-ENS are the two leading methods across variables and lead times. Similar to deterministic results, OmniCast performs worse than ECMWF-ENS at shorter lead times but outperforms this baseline beyond day 15.

## 5.2  OmniCast for medium-range forecasting

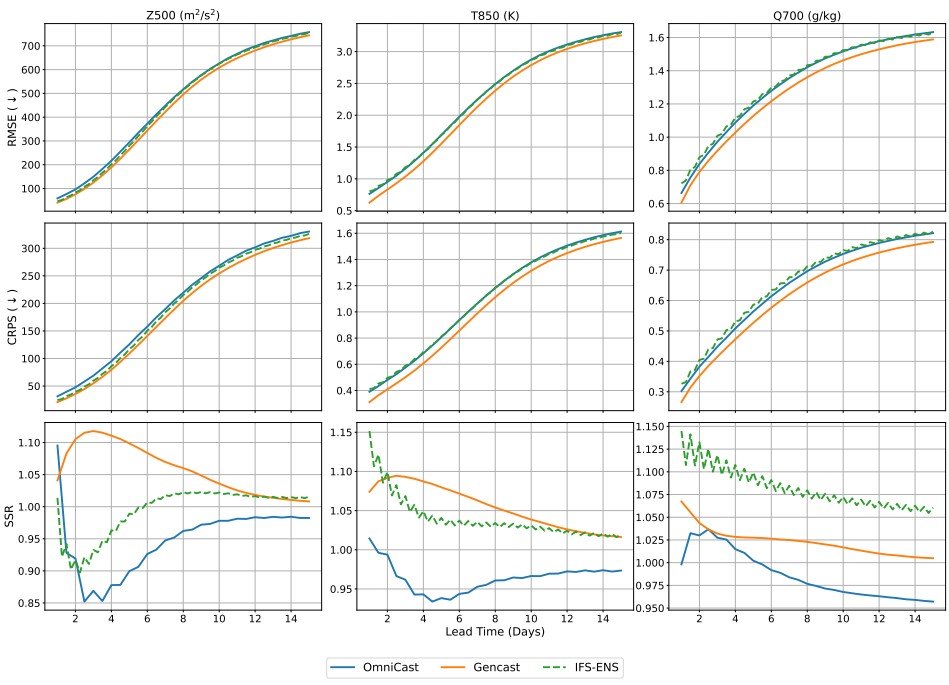

Figure 7: Probabilistic performance of different methods in medium-range forecasting. Solid curves are deep learning methods and dashed curves are numerical methods.

In addition to its strong performance on the S2S task, we demonstrate that OmniCast also performs competitively at the medium-range timescale. We train a VAE model with a spatial downsampling ratio of 16, compressing each weather state of shape $69 \times 721 \times 1440$ into a latent representation of size $256 \times 45 \times 90$. We then train OmniCast to predict two steps ahead at 12-hour intervals, following

the setup of Gencast [37]. During inference, we use autoregressive sampling, recursively feeding the most recent predicted frame as the new initial condition until the target lead time is reached. We generate forecasts using a single sampling iteration per frame with a diffusion temperature $\tau = 1.0$, and produce an ensemble of $50$ members.

We compare OmniCast against Gencast [37], a leading deep learning method for probabilistic forecasting, and IFS-ENS [23], the gold-standard numerical ensemble system. Following WeatherBench2, we use ensemble RMSE, CRPS, and spread-skill ratio (SSR) as evaluation metrics. As shown in Figure 7, OmniCast performs comparably with IFS-ENS across all variables and metrics, and is only slightly behind Gencast. These results indicate that OmniCast achieves strong performance across both medium-range and S2S timescales.

## 5.3 Efficiency of OmniCast

Beyond its strong empirical performance, OmniCast offers substantial efficiency gains over existing methods. We trained OmniCast for 4 days using 32 NVIDIA A100 GPUs. In comparison, Gencast requires 5 days of training on 32 TPUv5e devices – hardware significantly more powerful than A100s, and NeuralGCM [20] requires 10 days on 128 TPUv5e devices. Additionally, Gencast employs a two-stage training pipeline, first pretraining on $1.0°$ resolution and then finetuning on $0.25°$, while we trained OmniCast in a single stage.

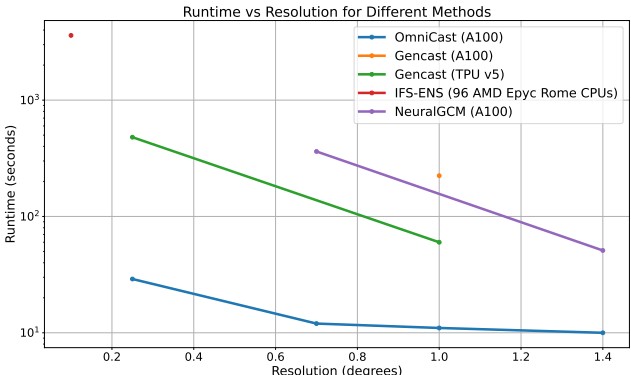

Figure 8: Runtime vs resolution to produce a 15-day forecast.

At inference time, OmniCast is orders of magnitude faster than Gencast, NeuralGCM, and IFS-ENS. Figure 8 compares the runtime (in seconds) required to generate a 15-day forecast across different resolutions. At $0.25°$ resolution, Gencast requires $480$ seconds on TPUv5, whereas OmniCast achieves the same forecast in just $29$ seconds on an A100. At $1.0°$, OmniCast completes inference in only $11$ seconds, compared to $224$ seconds for Gencast on the same hardware. These results highlight the scalability and practicality of OmniCast for operational forecasting.

The efficiency of OmniCast stems from two key architectural innovations. First, OmniCast operates in a much lower-dimensional latent space ($45 \times 90$ latent grid vs $721 \times 1440$ original grid), significantly reducing the computational cost of training and inference. Second, OmniCast employs a highly efficient sampling mechanism. Unlike Gencast, which performs 50 full forward passes through the entire network for 50 diffusion steps, OmniCast requires only a single forward pass through the transformer backbone. The subsequent diffusion steps involve only lightweight forward passes through a compact MLP diffusion head, resulting in orders-of-magnitude lower inference time. Together, these design choices enable OmniCast to deliver fast and scalable forecasts.

## 5.4 Ablation studies

We analyze four key factors that influence OmniCast's performance: the auxiliary deterministic objective, training sequence length $T$, unmasking order during sampling, and diffusion sampling temperature $\tau$. We present results for T850 on RMSE, CRPS, and SSR. We additionally study the impact of IC perturbations in Appendix B.3.

**Impact of the deterministic objective** Figure 9a demonstrates the important role of the deterministic loss in OmniCast's performance. Removing the MSE objective (No-MSE) degrades both RMSE and CRPS scores, with particularly noticeable impact at short lead times. However, naively applying MSE to all future frames (MSE-All-Frames) also proves counterproductive, as it forces deterministic predictions even for S2S timescales where weather systems become inherently chaotic. Our approach of applying MSE only to the first 10 frames achieves the best RMSE and CRPS scores across medium-range and S2S timescales.

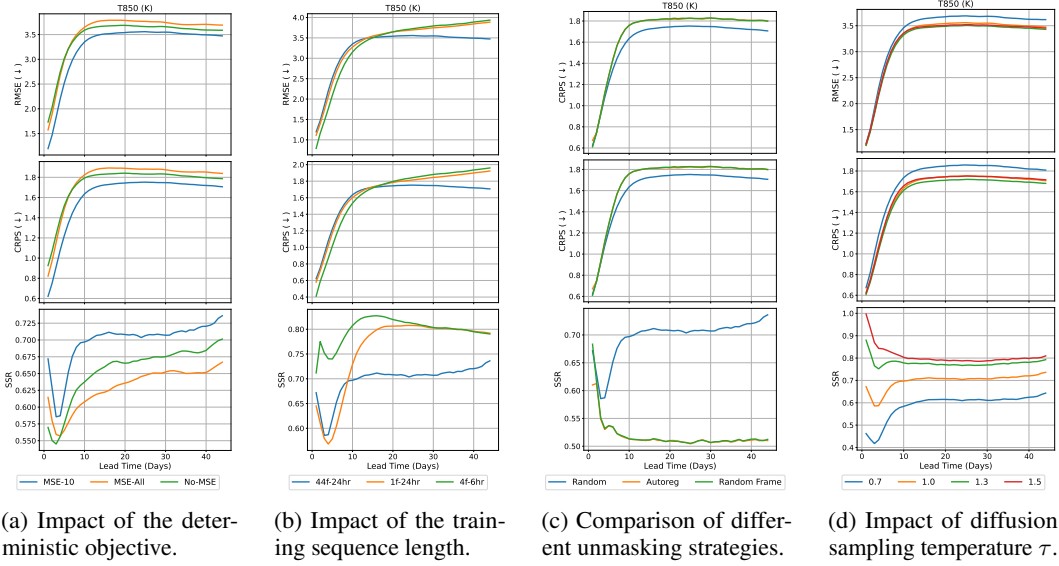

(a) Impact of the deterministic objective.

(b) Impact of the training sequence length.

(c) Comparison of different unmasking strategies.

(d) Impact of diffusion sampling temperature $\tau$.

Figure 9: Ablation studies showing the impact of different components in OmniCast.

**Impact of training sequence length** In the main S2S experiment, we train OmniCast to generate 44 future weather states at 24 hour intervals. One could alternatively train the model on shorter sequences and/or smaller intervals, then apply multiple roll-outs during inference to reach longer horizons. Figure 9b shows that models trained on shorter sequences or smaller intervals excel at short- and medium-range forecasting but underperform at S2S timescales. This trade-off emerges because shorter sequences allow models to specialize in near-term predictions, leading to better performance at shorter lead times. However, these models suffer from error accumulation at longer horizons, ultimately performing worse than the model trained on full sequences.

**Impact of unmasking orders** While our approach randomly masks tokens across both space and time during training, one may try more structured masking strategies at inference. We evaluate two such alternatives: an autoregressive strategy that unmasks entire frames sequentially, and a random framewise approach that unmasks complete frames in random order. Figure 9c shows that our fully randomized strategy achieves the best SSR scores, while both alternatives produce under-dispersive ensembles. The superior performance of the fully randomized approach stems from its introduction of additional randomness through the unmasking order, generating more diverse ensemble forecasts. This greater diversity consequently leads to better performance across other metrics.

**Impact of diffusion sampling temperature** Higher values of the temperature $\tau$ produce more diverse forecasts. Figure 9d demonstrates this empirically. Setting $\tau < 1$ produces under-dispersive ensembles, degrading performance across other metrics. Increasing $\tau$ boosts sample diversity, improving SSR scores and overall better performance. However, pushing $\tau$ too high (e.g., $\tau = 1.5$) causes samples to deviate from the mean prediction, compromising RMSE and CRPS performance. We identify $\tau = 1.3$ as the optimal value, providing the best balance between ensemble diversity and forecast quality, which we adopt for our main experiments.

## 6 Conclusion

We present OmniCast, a novel latent diffusion model for S2S prediction. By combining the masked generative framework with a diffusion objective, our approach enables direct modeling of long sequences of future weather states while avoiding error accumulation inherent in autoregressive methods. OmniCast achieves state-of-the-art performance in deterministic and probabilistic metrics while maintaining exceptional physical consistency. In medium-range forecasting, OmniCast performs competitively with existing methods while being significantly more efficient. Future work could study the fundamental trade-off between VAE reconstruction quality and transformer modeling capacity, and explore more sophisticated generative frameworks to enhance the diffusion objective.

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

# A  Implementation details

## A.1  VAE details

Our VAE model follows the UNet implementation from PDEArena [12]. We use the following hyperparameters for UNet in our experiments.

Table 1: Default hyperparameters of UNet

| Hyperparameter | Meaning | Value |
|---|---|---|
| Padding size | Padding size of each convolution layer | 1 |
| Kernel size | Kernel size of each convolution layer | 3 |
| Stride | Stride of each convolution layer | 1 |
| Input channels | The number of channels of the input | 69 |
| Input channels | The number of channels of the output | 69 |
| Base channels | The base hidden dimension of the UNet | 256 |
| Channel multiplications | Determine the number of output channels for Down and Up blocks | $[1, 2, 4, 4, 8]$ |
| Dimension of $z$ | The dimension of the latent space | 1024 |
| Blocks | Number of blocks | 2 |
| Use attention | If use attention in Down and Up blocks | False |
| Dropout | Dropout rate | 0.0 |

The VAE encoder embeds each weather state of shape $69 \times 128 \times 256$ to a latent map of shape $1024 \times 8 \times 16$, reducing the spatial dimensions by 16. We use a KL weight of $5e - 5$ and optimize the VAE model with Adam [18] for 200 epochs with a batch size of 32, a base learning rate of $2e - 4$, parameters ($\beta_1 = 0.9, \beta_2 = 0.95$), and weight decay of $1e - 5$. The learning rate follows a linear warmup for the first 20 epochs, followed by a cosine decay schedule for the remaining 180 epochs.

## A.2  Weighted deterministic objective

In OmniCast, we employ a weighted MSE objective to encourage accurate deterministic predictions for near-term frames. The objective is formulated as:

$$\mathcal{L}_{\text{deter}}(\theta) = \mathop{\mathbb{E}}_{\mathbf{m} \sim p_{\mathcal{U}}} \left[ \sum_{m_i = 1} w(i) \| x_i - \hat{x}_i \|_2^2 \right], \tag{6}$$

where $w(i)$ is an exponentially decreasing weighting function. We compute this weight in three steps. First, for each token $i$, we determine its corresponding frame index $k = \lfloor \frac{i}{h \times w} \rfloor$, where $h \times w$ represents the spatial dimensions of each frame's latent map. Second, we assign weights to tokens based on their frame index: $w(i) = e^{-k} = e^{-\lfloor \frac{i}{h \times w} \rfloor}$, ensuring all tokens from the same frame receive equal weight. Third, we set $w(i) = 0$ for tokens beyond frame 10 and normalize the remaining weights to sum to one.

## A.3  Optimization details

We optimize OmniCast with AdamW [18] for 100 epochs with a batch size of 32, a base learning rate of $2e - 4$, parameters ($\beta_1 = 0.9, \beta_2 = 0.95$), and weight decay of $1e - 5$. The learning rate follows a linear warmup for the first 10 epochs, followed by a cosine decay schedule for the remaining 90 epochs.

# B  Additional experiments

## B.1  VAE reconstruction quality

The VAE model is a critical component in OmniCast, since it imposes an upper bound on the forecasting performance. We dedicated substantial efforts to designing the VAE model that balances

reconstruction quality and compression. Our primary goal was to achieve a high compression ratio along the spatial dimensions, as this directly reduces the number of training tokens required for the subsequent transformer model. We employed $16\times$ spatial reduction for both the medium-range setting with $0.25°$ data and the S2S setting with the $1.40625°$ data. We then incrementally increased the latent dimension until we obtained an acceptable reconstruction error. Table 2 shows that increasing the latent dimension consistently improves the reconstruction errors across different variables. We did not increase the latent dimension beyond $1024$ since it would create difficulties for training with the diffusion objective. It is also noticeable that the VAE model trained on $0.25°$ data performs much better than the one trained on $1.40625°$ with the same spatial compression ratio and latent dimension. This is expected since higher-resolution data has more spatial redundancy, leading to easier compression for the VAE model. Figures 10 and 11 visualize the VAE reconstructions for 6 surface and pressure-level variables. The VAE was able to retain important details and structures of the input, albeit slightly smoothing out the data.

Table 2: **Reconstruction error of VAE models for different physical variables and latent dimensions** ($D$). Results are shown for datasets at two spatial resolutions: $1.40625°$ (left) and $0.25°$ (right). Lower values indicate better reconstruction.

| | $1.40625°$ resolution | | | | | | $0.25°$ resolution | | | | |
| --- | --- | --- | --- | --- | --- | --- | --- | --- | --- | --- | --- |
| | T2m | U10 | V10 | Z500 | T850 | | T2m | U10 | V10 | Z500 | T850 |
| $D = 256$ | 0.96 | 0.65 | 0.62 | 48.72 | 0.77 | | **0.55** | **0.25** | **0.23** | **18.77** | **0.37** |
| $D = 512$ | 0.80 | 0.51 | 0.48 | 35.42 | 0.64 | | — | — | — | — | — |
| $D = 1024$ | **0.71** | **0.43** | **0.40** | **27.34** | **0.57** | | — | — | — | — | — |

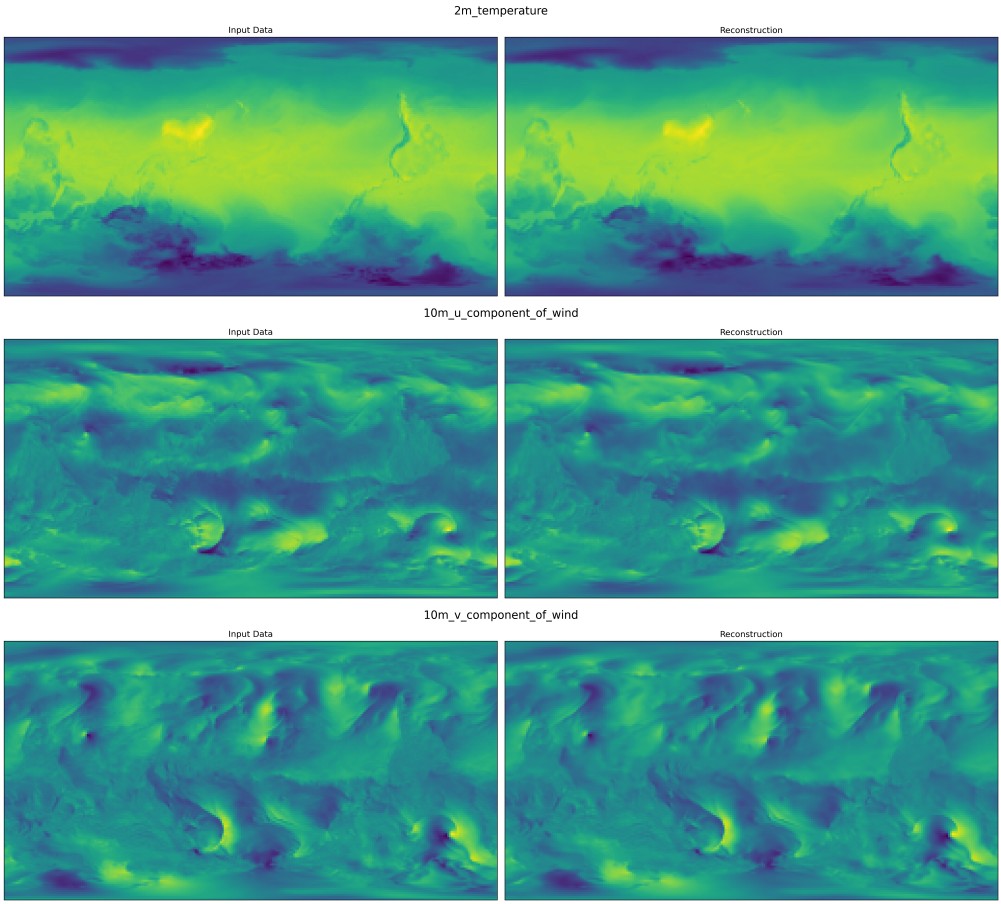

Figure 10: Reconstructions of the VAE model for T2m, U10, and V10.

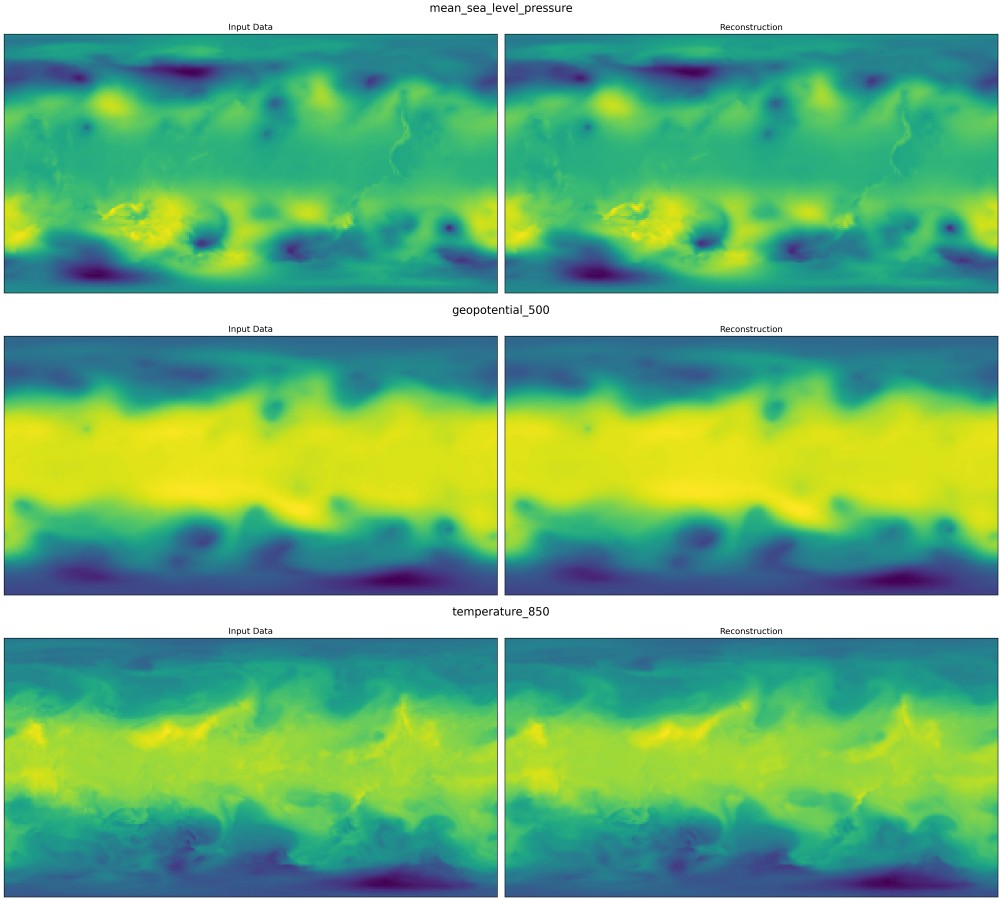

Figure 11: Reconstructions of the VAE model for MSLP, Z500, and T850.

Before selecting the specific VAE model presented in our paper, we also tried two alternative VAE architectures: VQ-VAE [45], which compresses data into a discrete latent space, and Video VAE [1], which compresses data across both spatial and temporal dimensions. Our early experiments with VQ-VAE did not achieve satisfactory reconstruction qualities, as the errors were consistently 2 to 3 times higher than those obtained with a continuous VAE using the same spatial downsampling factor. We also found that for an equivalent effective compression ratio, a per-frame VAE consistently outperformed a video VAE. These results led us to opt for the per-frame continuous VAE model.

## B.2 Comparison with more deep learning baselines

In addition to PanguWeather and GraphCast, we compare OmniCast with two advanced transformer-based methods: ClimaX [32] and Stormer [33]. Figure 12 shows that Stormer achieves superior accuracy in short-to-medium timescales, consistent with its reported results. However, as an autoregressive method, its performance degrades more rapidly than OmniCast, eventually falling below Climatology, albeit at a slower rate than PanguWeather and GraphCast. ClimaX takes a different approach as a direct forecasting method, where a model trained on large-scale climate data is finetuned specifically for individual lead times. This approach avoids error accumulation and achieves comparable performance with OmniCast at S2S scales. However, ClimaX requires fine-tuning separate models for each target lead time, while a single OmniCast model can simultaneously generate the complete sequence of future weather states.

## B.3 Impact of IC perturbations

Initial condition (IC) perturbations—adding random noise to initial conditions $X_0$ – are a standard technique in numerical methods for generating ensemble forecasts. This approach complements our

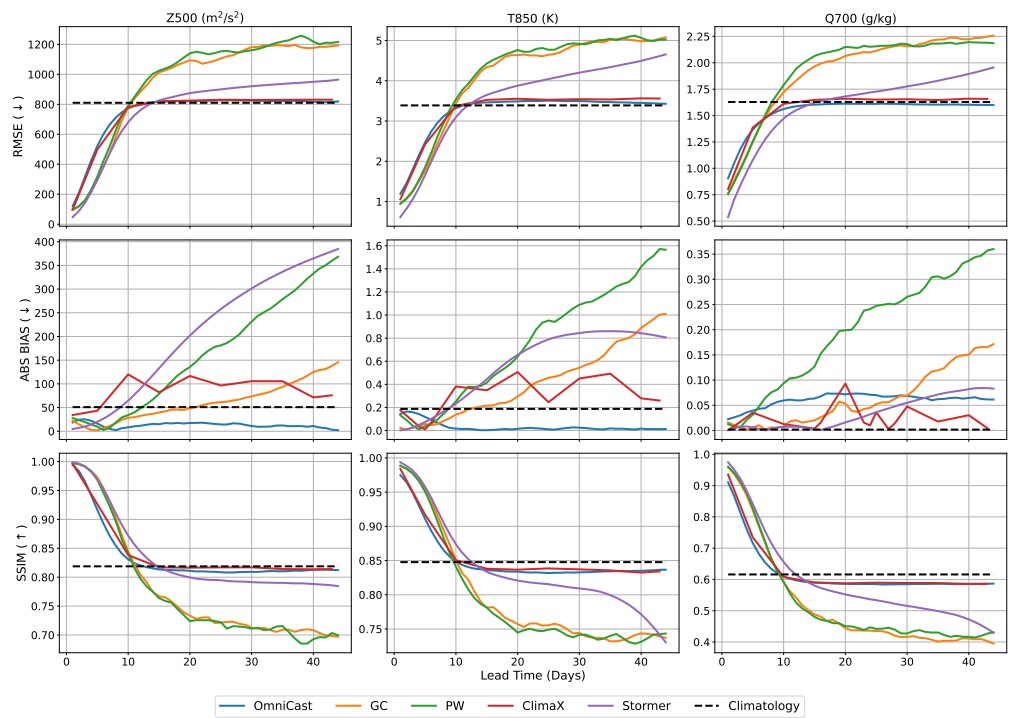

Figure 12: Comparison of deterministic performance of OmniCast with more deep learning methods.

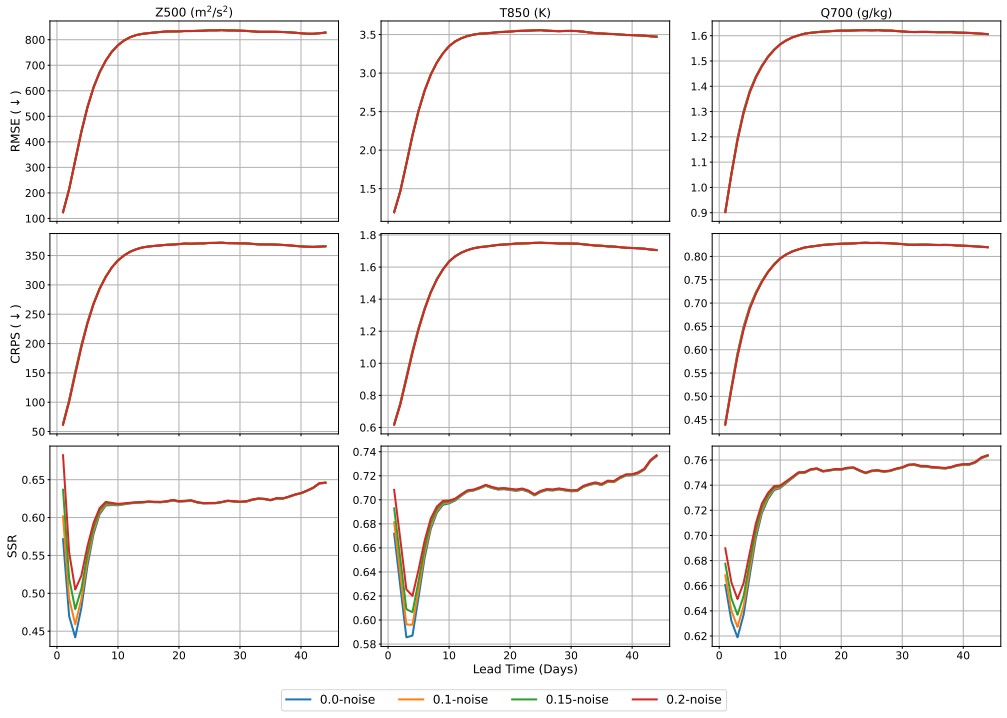

Figure 13: Performance of OmniCast with different levels of IC noise.

generative framework. Figure 13 evaluates OmniCast's performance across different noise levels, varying the standard deviation of the Gaussian distribution used for generating perturbations. The results demonstrate OmniCast's robustness to input noise, maintaining consistent RMSE and CRPS

scores across noise levels from 0.0 to 0.2, with only minor variations in SSR scores at short lead times.

## B.4    Scaling inference compute

Finally, we examine how increasing inference compute affects OmniCast's performance through two hyperparameters: the number of ensemble forecasts and the average number of unmasking iterations per frame, i.e., 1-iter means a total of 44 iterations for 44 frames. Figure 14 shows that generating more ensemble forecasts improves both system diversity (higher SSR) and mean prediction accuracy (lower RMSE). Interestingly, while increasing the number of unmasking iterations shows minimal impact on RMSE, it yields slight improvements in SSR. This improvement likely stems from the increased randomness in unmasking order with more iterations, leading to greater ensemble diversity.

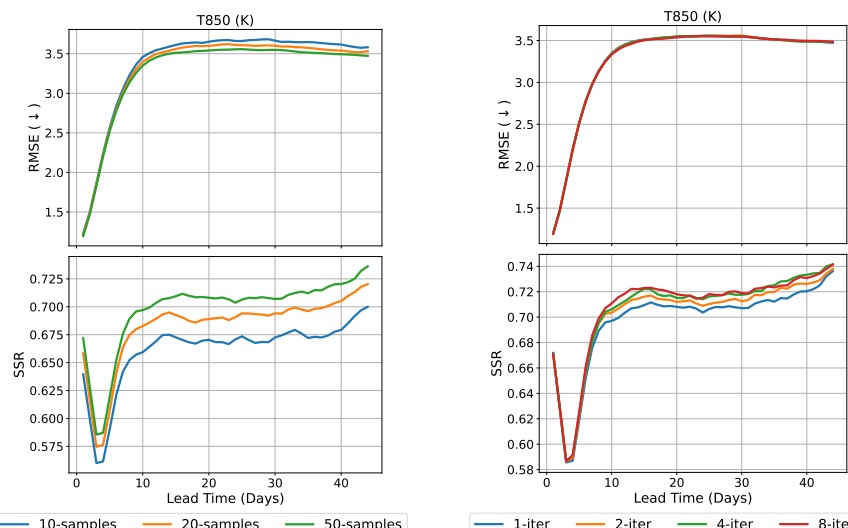

Figure 14: Performance of OmniCast as we vary the number of ensemble forecasts (left) and the number of unmasking iterations.

## B.5    Testing OmniCast stability

We tested the stability of OmniCast by rolling out the model to 100 years into the future. We found that OmniCast consistently produces stable and physically feasible forecasts, even at 100 years ahead. Please see below for visualizations of OmniCast's rollouts for various weather variables with 4 samples each.

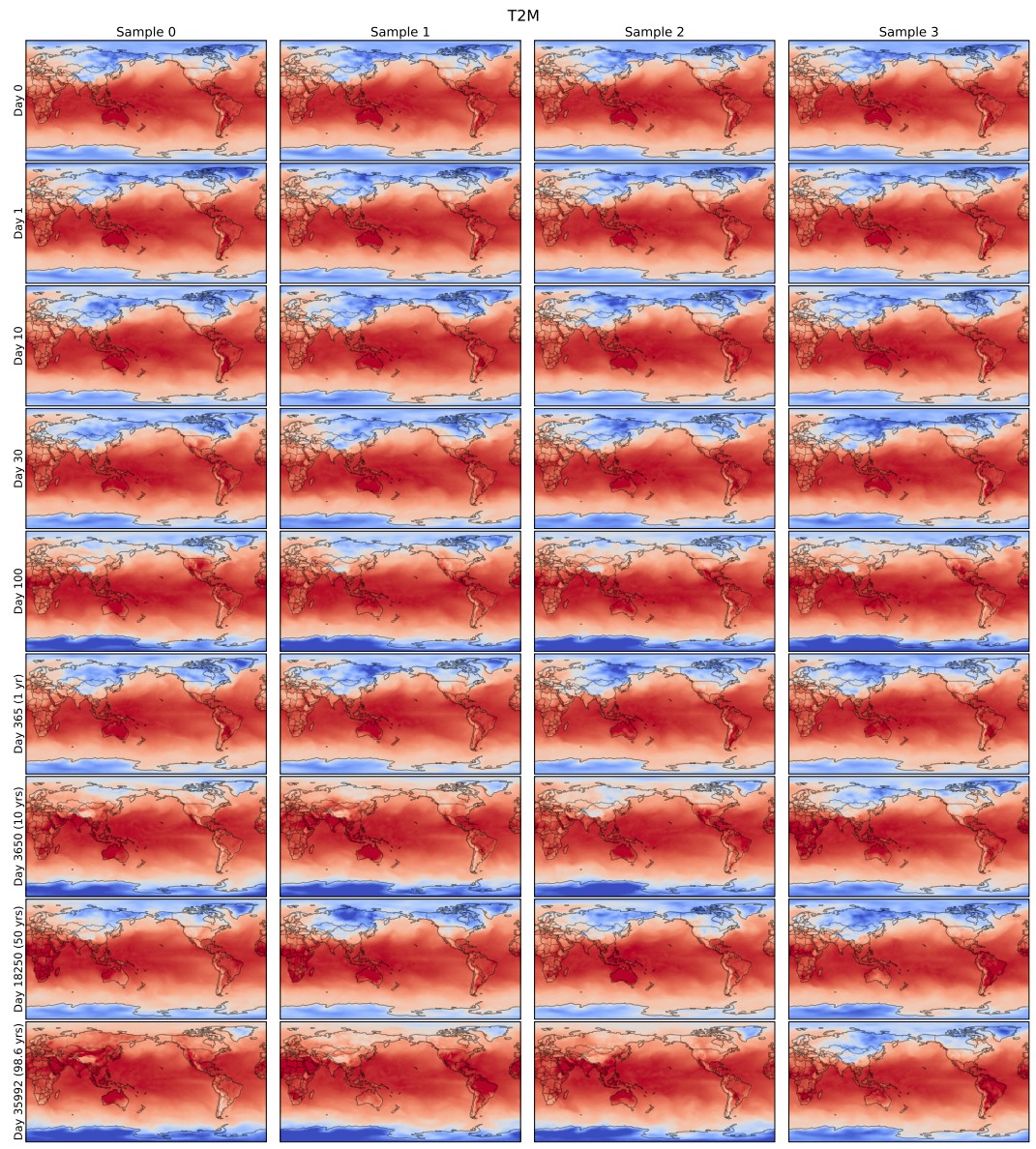

Figure 15: Rollouts for 2-meter temperature up to 100 years ahead.

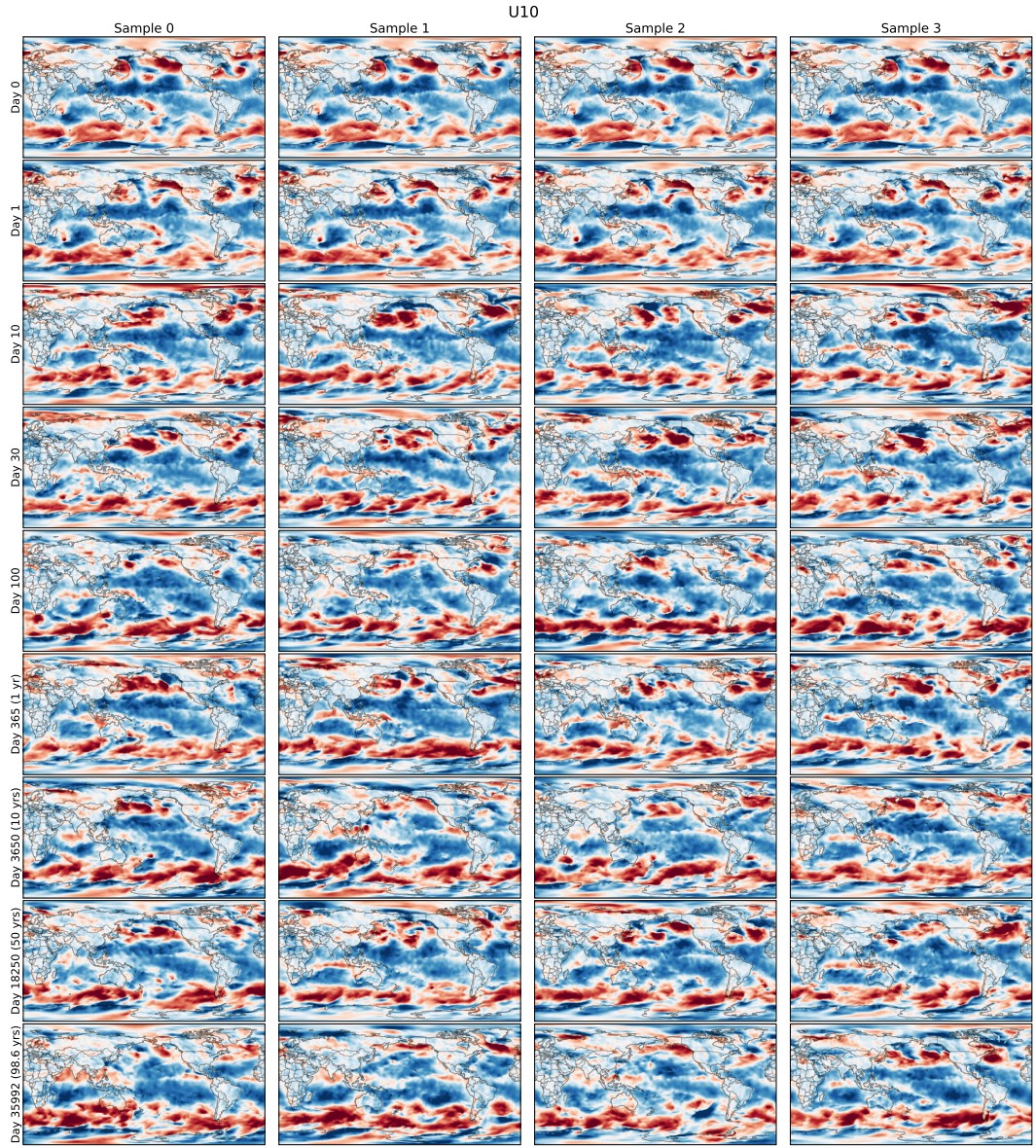

Figure 16: Rollouts for 10-meter u component of wind up to 100 years ahead.

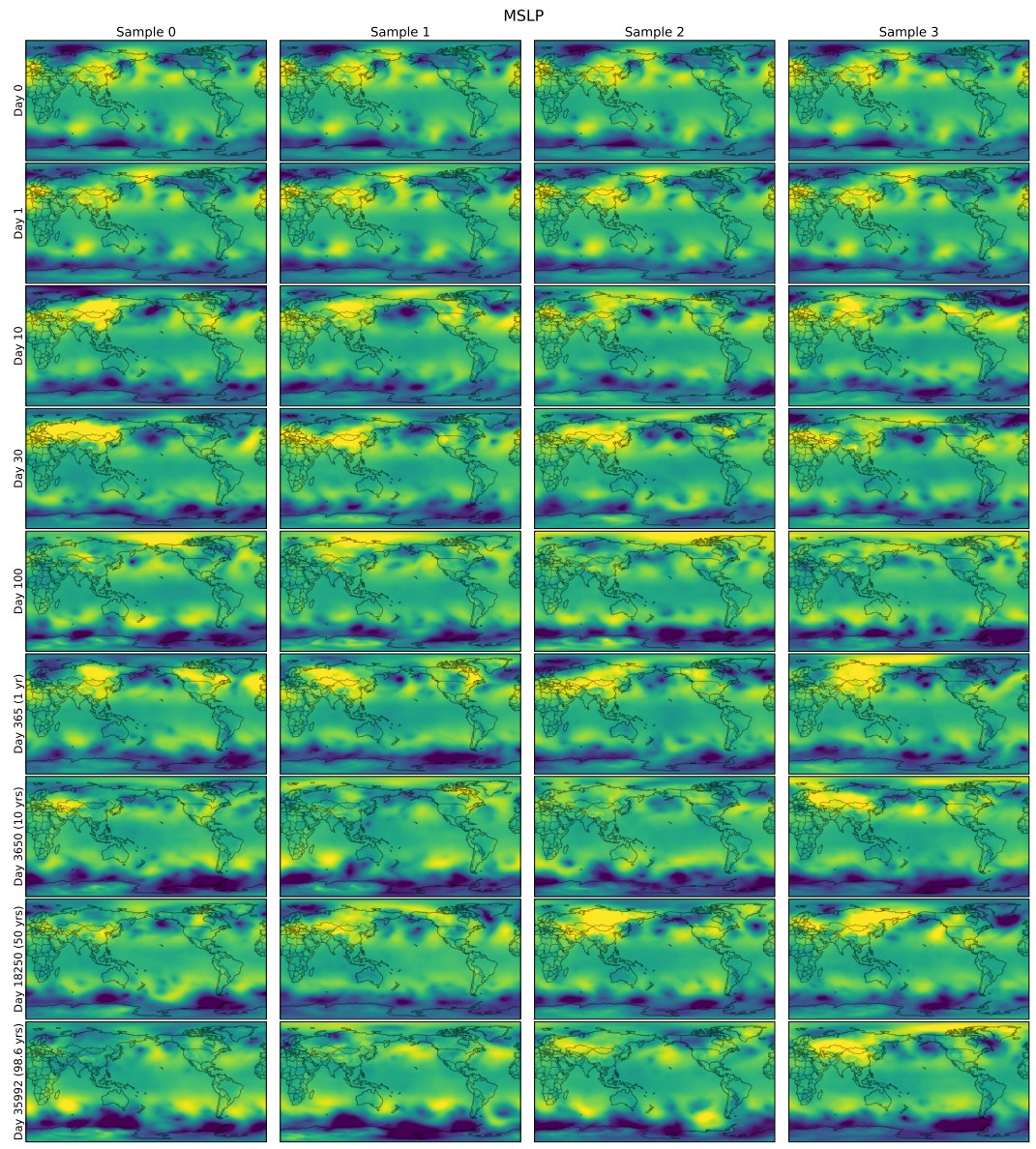

Figure 17: Rollouts for mean sea level pressure up to 100 years ahead.

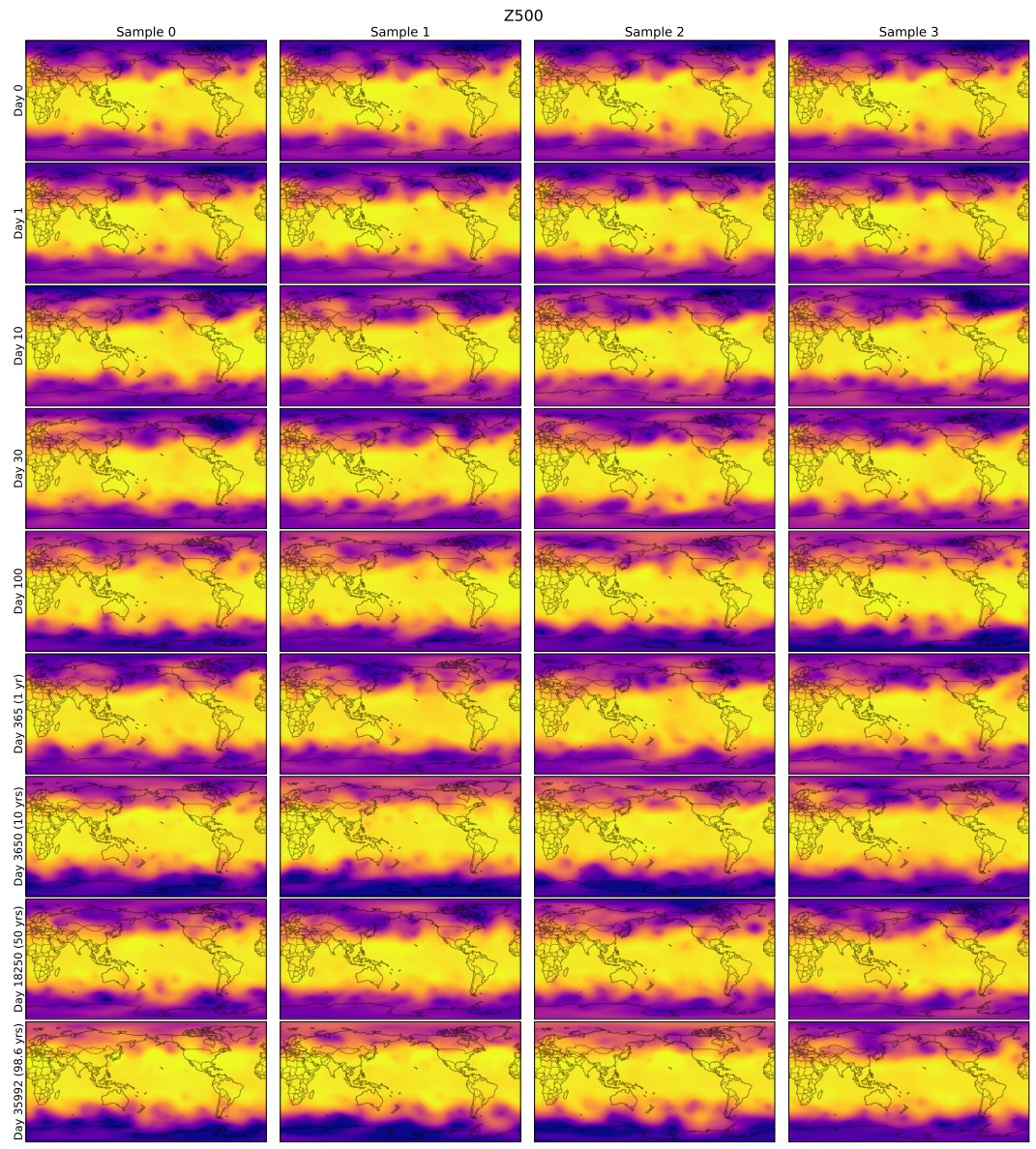

Figure 18: Rollouts for 500hPa geopotential up to 100 years ahead.

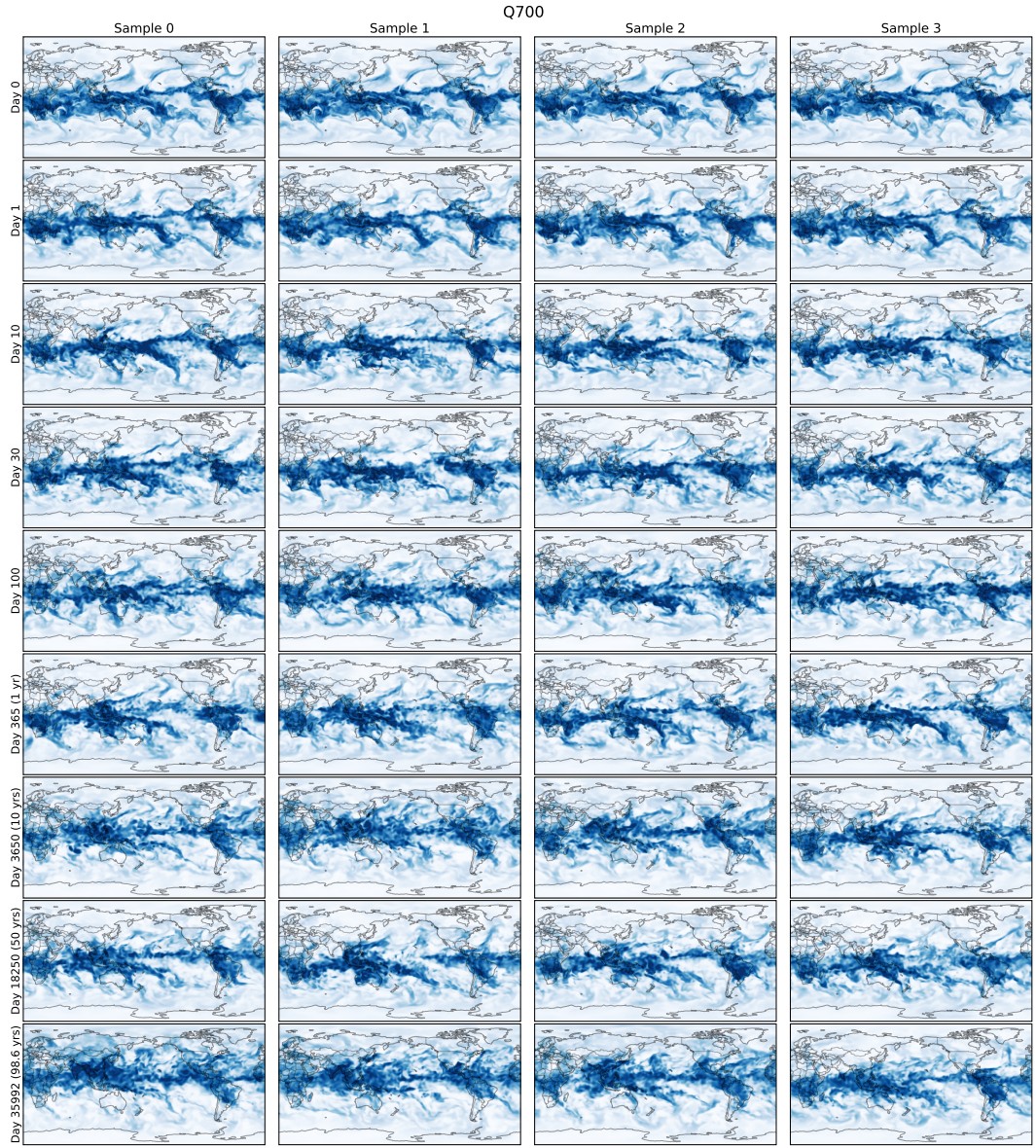

Figure 19: Rollouts for 700hPa specific humidity up to 100 years ahead.

