# OpenReview forum: "OmniCast: A Masked Latent Diffusion Model for Weather Forecasting Across Time Scales"
_NeurIPS.cc/2025/Conference — NeurIPS 2025 poster_

### Official Review · Reviewer_Er6Q · 2025-06-30

**Clarity:** 3
**Significance:** 2
**Originality:** 3
**Rating:** 3
**Confidence:** 4

**Summary:**

This paper introduces SeasonCast, a novel latent diffusion model tailored for subseasonal-to-seasonal (S2S) weather prediction, targeting lead times of 2 to 6 weeks. Unlike traditional autoregressive methods that often struggle with error accumulation over longer time horizons, SeasonCast takes a two-stage approach to address this challenge:
	1	A Variational Autoencoder (VAE) compresses high-dimensional weather data into a smooth, lower-dimensional latent space.
	2	A masked transformer with a diffusion head then generates future latent tokens iteratively, conditioned on both the initial state and partially revealed future tokens.
This design allows SeasonCast to jointly model spatial and temporal dependencies, effectively reducing compounding errors. It also enables the model to learn both short-term dynamics (initial conditions) and long-term patterns (boundary conditions).
Experimental results show that SeasonCast not only outperforms deep learning baselines like Pangu-Weather and GraphCast, but also rivals numerical ensemble models such as ECMWF. It delivers strong results across deterministic (RMSE, SSIM), physics-based (SDIV, SRES), and probabilistic metrics at S2S scales—while also demonstrating competitive performance in medium-range forecasting (1–15 days), all with minimal computational overhead.

**Questions:**

1. Clarify Initial and Boundary Conditions I recommend that the authors provide a brief explanation or context when referring to initial conditions and boundary conditions. This would be especially helpful for readers who may not have a background in climate science or differential equations, allowing them to better understand the modeling framework and its assumptions.

2. Include Surface-Level Variable Performance It would also strengthen the paper to report performance on key surface-level variables, such as 2-meter temperature (T2m) and 10-meter wind components (U10, V10). These variables are critical for many practical applications of S2S forecasting and would offer a more complete picture of the model’s real-world utility.

**Ethical Concerns:**

["NO or VERY MINOR ethics concerns only"]

**Final Justification:**

Thank you to the authors for their time and effect during the rebuttal. I still think Climatology is a very crucial benchmark in S2S forecasting, and there are prior works showing ML models can outperform Climatology at S2S scale. For this reason,  I would like to  keep my original score.

**Limitations:**

Yes

**Quality:**

3

**Strengths And Weaknesses:**

The paper presents a thoughtfully designed two-stage framework, SeasonCast, which combines a VAE with a masked transformer and diffusion head to tackle the persistent challenge of error accumulation in autoregressive S2S forecasting. The use of latent diffusion for probabilistic modeling is a particularly clever and well-justified choice.

The model is rigorously evaluated on both ChaosBench and WeatherBench2, with comparisons against leading deep learning models (Pangu-Weather, GraphCast) and numerical baselines (ECMWF-ENS). The evaluation spans a wide range of metrics, including deterministic (RMSE, SSIM), physics-based (SDIV, SRES), and probabilistic (CRPS, SSR) measures.

However, I do have some concerns about SeasonCast’s performance at S2S scales. Looking at Figure 4, the model often performs on par with—or even worse than—climatology across most variables and metrics, with the exception of absolute bias for Z500 and T850. This raises concerns about its readiness for operational use. Additionally, the paper only presents SeasonCast’s performance on a subset of atmospheric variables, leaving out key surface-level indicators like 2-meter temperature and 10-meter wind components (U and V). Including these would provide a more complete picture of the model’s practical utility.

---

> ### Author Rebuttal · Authors · 2025-07-30
>
> We thank the reviewer for the constructive feedback and for recognizing the design, rigorous evaluation, and strong performance of SeasonCast. We hope to address the reviewer’s main questions and concerns below.
>
> > Looking at Figure 4, the model often performs on par with—or even worse than—climatology across most variables and metrics, with the exception of absolute bias for Z500 and T850.
>
> Theoretical limits in deterministic forecasting state that no model can surpass climatology beyond 14 days. **However, S2S prediction is not purely about deterministic skill; robust probabilistic uncertainty estimation is crucial.**
> - Climatology inherently lacks uncertainty estimates, rendering it impractical for real-world forecasting applications where uncertainty is important for informed decision-making.
> - SeasonCast demonstrably improves upon climatology by providing reliable probabilistic forecasts, which are far more valuable for decision-making in risk-sensitive applications.
>
> **Across all benchmarks and metrics, SeasonCast consistently produces stable, physically consistent, and well-calibrated forecasts for S2S.** While achieving comparable RMSE due to theoretical limits, SeasonCast offers significant benefits over climatology through its superior probabilistic forecasting capabilities.
>
> > Additionally, the paper only presents SeasonCast’s performance on a subset of atmospheric variables, leaving out key surface-level indicators like 2-meter temperature and 10-meter wind components (U and V).
>
> We evaluated SeasonCast on a total of 69 different weather variables, but were not able to show all of them due to the page limits. **Please see below the performance of SeasonCast across different key surface-level variables and various metrics.** We will also update the Appendix to include this result.
>
> | Lead time (days) | RMSE (T2M) | RMSE (U10) | RMSE (V10) | CRPS (T2M) | CRPS (U10) | CRPS (V10) | SSR (T2M) | SSR (U10) | SSR (V10) |
> |------------------|------------|------------|------------|------------|------------|------------|-----------|-----------|-----------|
> | 25               | 2.59       | 3.87       | 4.00       | 1.11       | 1.88       | 1.94       | 0.69      | 0.83      | 0.85      |
> | 35               | 2.56       | 3.85       | 3.98       | 1.10       | 1.87       | 1.93       | 0.69      | 0.83      | 0.85      |
> | 40               | 2.56       | 3.84       | 3.98       | 1.10       | 1.87       | 1.92       | 0.70      | 0.83      | 0.85      |
>
> > I recommend that the authors provide a brief explanation or context when referring to initial conditions and boundary conditions
>
> We fully agree with the reviewer and will update the paper to discuss the initial conditions and boundary conditions problems in the future version of the paper.

---

> > ### Author Response · Authors · 2025-08-04
> > **Reminder of our rebuttal**
> >
> > Dear Reviewer Er6Q,
> >
> > We are writing to follow up on our rebuttal, in which we have done our best to address your concerns with additional experiments and detailed clarifications. As the discussion period closes in two days, we sincerely hope you have a moment to review our response. We are eager to address any remaining questions you might have and would be happy to discuss further. Thank you again for your valuable feedback.
> >
> > Best regards,
> >
> > The Authors

---

> > > ### Author Response · Authors · 2025-08-06
> > > **Reminder of our rebuttal**
> > >
> > > Dear Reviewer Er6Q,
> > >
> > > We are writing to follow up on our rebuttal, in which we have done our best to address your concerns with additional experiments and detailed clarifications. As the discussion period is coming to an end soon, we sincerely hope you have a moment to review our response. We are eager to address any remaining questions you might have and would be happy to discuss further. Thank you again for your valuable feedback.
> > >
> > > Best regards,
> > >
> > > The Authors

---

### Official Review · Reviewer_pne4 · 2025-07-01

**Clarity:** 3
**Significance:** 3
**Originality:** 3
**Rating:** 5
**Confidence:** 4

**Summary:**

This paper proposes a novel model named SeasonCast for subseasonal-to-seasonal (S2S) scale weather forecasting. By combining a masked Transformer model with diffusion models to directly model complete sequences of future weather states in a low-dimensional latent space, SeasonCast effectively avoids the error accumulation problems inherent in numerical weather prediction (NWP) and traditional autoregressive approaches. Experimental results demonstrate that SeasonCast achieves state-of-the-art performance at S2S scales while remaining competitive for medium-range forecasting, with higher computational efficiency than existing methods.

**Questions:**

1. Could you provide visual examples of forecasts compared with observations, especially for extreme events? This would help readers better understand the spatial and temporal characteristics your model captures.

2. Have you evaluated SeasonCast’s skill using standard meteorological correlation metrics like ACC and ACT? If not, would you consider adding these to better align with operational forecast verification practice?

**Ethical Concerns:**

["NO or VERY MINOR ethics concerns only"]

**Final Justification:**

Authors explained the absence of qualitative figures as an oversight and committed to adding them. The effective architecture, comprehensive experiments, and strong long-lead performance together justify acceptance. While the novelty is somewhat incremental, I believe its solid practical effectiveness warrants acceptance.

**Limitations:**

Yes.

**Paper Formatting Concerns:**

N/A.

**Quality:**

3

**Strengths And Weaknesses:**

#### Strengths:

1. **Clear motivation and solid problem formulation.** The paper provides a thorough discussion of the challenges in S2S forecasting, particularly the limitations of autoregressive architectures in modeling both initial and boundary conditions at long lead times. The proposed masked generative framework is well-motivated as a natural solution to address these issues.
2. **Innovative architecture design.** The combination of a continuous VAE to compress high-dimensional weather states, with a Transformer trained under a masked latent diffusion objective, is novel for weather forecasting and technically sound.
3. **Substantial research effort.** The experimental design is comprehensive, with thorough comparisons to multiple baselines, including leading numerical weather prediction systems (e.g., ECMWF, UKMO, and CMA) and recent AI models like PanguWeather and GraphCast. This is particularly valuable given that some NWP outputs (such as CMA) are not easily accessible, showing the authors’ strong commitment to robust benchmarking.
3. **Well-structured presentation.** The methodology is described in detail and is well-structured, making the proposed approach easy to follow and the experiments reproducible in principle.
4. **Strong results.** The model achieves state-of-the-art performance at subseasonal-to-seasonal scales and remains competitive at medium-range timescales, with notable computational efficiency improvements.

#### Weaknesses:

1. **Limited interpretability and case studies.** While the paper includes extensive quantitative benchmarks, it would benefit from more qualitative analyses or illustrative case studies (e.g., maps of extreme events, typical forecast snapshots) to show when and why SeasonCast outperforms or underperforms compared to other systems.
2. **Missing some common meteorological verification metrics.** Although the paper uses several meaningful metrics, it does not report widely adopted indicators such as the Anomaly Correlation Coefficient (ACC) or Forecast Activity (ACT), which are standard in meteorological skill assessments. Including these would strengthen the claim of operational relevance.

---

> ### Author Rebuttal · Authors · 2025-07-30
>
> We thank the reviewer for the constructive feedback and for recognizing the contributions, novel architecture design, presentation, and strong performance of SeasonCast. We hope to address the reviewer’s main questions and concerns below.
>
> > While the paper includes extensive quantitative benchmarks, it would benefit from more qualitative analyses or illustrative case studies.
>
> We agree with the reviewer that additional illustrative studies would enhance our understanding of SeasonCast's performance. As the rebuttal phase prohibits uploading PDF files or linking to external pages, we will incorporate VAE reconstructions, sample forecasts, and extreme weather events into a future version of the paper.
>
> > Although the paper uses several meaningful metrics, it does not report widely adopted indicators such as the Anomaly Correlation Coefficient (ACC) or Forecast Activity (ACT), which are standard in meteorological skill assessments.
>
> We further evaluated SeasonCast’s performance on the Anomaly Correlation Coefficient (ACC) metric, which we summarized in the table below. We left out Z500 because there was a bug in the climatology data provided by ChaosBench. We have contacted the authors to have this fixed and will report the results on more variables in the updated version of the paper.
>
> | Lead time (days) | SeasonCast (T850) | SeasonCast (Q700) | ECMWF-ENS (T850) | ECMWF-ENS (Q700) |
> |------------------|-------------------|-------------------|-------------------|-------------------|
> | 25               | 0.08              | 0.09              | 0.08              | 0.13              |
> | 35               | 0.08              | 0.08              | 0.06              | 0.10              |
> | 40               | 0.08              | 0.08              | 0.06              | 0.09              |
>
> This result demonstrates that SeasonCast performs competitively with ECMWF-ENS at S2S timescales in terms of the ACC metric and also maintains stable performance even up to day 40, reinforcing the conclusion observed across other metrics presented in our paper.

---

> > ### Comment · Reviewer_pne4 · 2025-08-01
> >
> > I understand that practical constraints may have limited your ability to provide a more comprehensive response, but the current rebuttal contains insufficient information to fully address my concerns. Therefore, I would like to follow up with additional questions:
> >
> > 1. Could you briefly explain the rationale behind not including any figures in the original paper? What considerations led to this decision?
> >
> > 2. I find it unclear why some variables were available while Z500—such a critical variable—was not. Could you clarify this discrepancy? Additionally, what about the Forecast Activity (ACT) metric? Was this evaluated, and if not, why?

---

> > > ### Author Response · Authors · 2025-08-01
> > >
> > > We thank the reviewer for the quick follow-up and the opportunity to provide a more detailed clarification.
> > >
> > > 1. Regarding Qualitative Figures. We sincerely apologize for the lack of qualitative figures; **it was an honest oversight and not an intentional omission.** Our primary focus was on conducting the extensive quantitative comparisons and analyses needed for a robust evaluation, and in managing these numerous experiments, we simply neglected to include forecast snapshots. We promise to add the following to the final version: VAE reconstructions of the weather states, sample forecasts for both S2S and medium-range settings, and case studies of extreme weather events. We hope the current breadth of comparative experiments is sufficient to demonstrate the strong performance of our method in the meantime.
> > >
> > > 2. Regarding Z500 ACC and the ACT Metric. **We have fixed the issue with Z500 ACC.** After contacting the ChaosBench authors, we confirmed it was a unit discrepancy between the climatology (which used geopotential height, $m/s^2$) and the data (which used geopotential, $m^2/s^2$). The table below reports ACC across key variables, including Z500 after the fix.
> > >
> > > | Lead time (days) | SeasonCast (Z500) | SeasonCast (T850) | SeasonCast (Q700) | ECMWF-ENS (Z500) | ECMWF-ENS (T850) | ECMWF-ENS (Q700) |
> > > |------------------|-------------------|--------------------|--------------------|-------------------|--------------------|--------------------|
> > > | 25               | 0.13              | 0.08               | 0.09               | 0.08              | 0.08               | 0.13               |
> > > | 35               | 0.12              | 0.08               | 0.08               | 0.07              | 0.06               | 0.10               |
> > > | 40               | 0.12              | 0.08               | 0.08               | 0.07              | 0.06               | 0.09               |
> > >
> > > **We did not report the Forecast Activity (ACT) metric as it is not a common metric in standard AI weather benchmarks (e.g., WeatherBench2, ChaosBench).** We believe our paper already provides a very thorough evaluation using a comprehensive suite of eight metrics: RMSE, ACC, Absolute BIAS, and SSIM for accuracy, SDIV and SRES as physics-based metrics, and CRPS and SSR as probabilistic metrics.
> > >
> > > Nevertheless, we are happy to add ACT if you believe it is crucial. **We were unable to find a standard definition in our search; if you could point us to a canonical reference, we would gladly compute it and report it.**
> > >
> > > We hope this fully addresses your concerns and thank you again for your constructive feedback.

---

> > > > ### Comment · Reviewer_pne4 · 2025-08-01
> > > >
> > > > Thank you for the reply! I' ll keep my score.

---

> > > > > ### Author Response · Authors · 2025-08-04
> > > > > **Thank you**
> > > > >
> > > > > We thank the reviewer again for the constructive feedback, which we believe will significantly improve our paper.

---

### Official Review · Reviewer_SUwF · 2025-07-04

**Clarity:** 2
**Significance:** 3
**Originality:** 2
**Rating:** 4
**Confidence:** 4

**Summary:**

This paper proposes SeasonCast for S2S prediction, which consists of two components, a VAE model that encodes raw weather data into a continuous, lower-dimensional latent space, and a diffusion-based transformer model that generates a sequence of future latent tokens given the initial conditioning tokens. It achieves SOTA accuracy and efficiency. While experimentally thorough, it lacks physics constraints and extreme-event validation.

**Questions:**

1. How does SeasonCast enforce physical laws (e.g., mass conservation) in generated forecasts? Are there constraints in the loss function or latent space?
2. What is the per-variable reconstruction error (e.g., RMSE) for the VAE? Does compression degrade key variables like precipitation?
3. Can SeasonCast predict rare events (e.g., heatwaves)? Please provide case studies on ChaosBench extreme indices.
4. How does performance degrade beyond 44 days? Is there a plan to support flexible forecasting windows?

**Ethical Concerns:**

["NO or VERY MINOR ethics concerns only"]

**Final Justification:**

Thank you for the author's rebuttal, which solved my questions. I will increase my review score.

**Quality:**

3

**Strengths And Weaknesses:**

strength
1. Masked diffusion transformer applies diffusion loss for probabilistic forecasting + auxiliary MSE loss (weighted for near-term frames) for deterministic accuracy.
2. Masked diffusion generates full future sequences (44 days) in a single forward pass, eliminating autoregressive error accumulation and capturing boundary conditions.
3. Outperforms deep learning (DL) baselines (PanguWeather, GraphCast) and matches ECMWF-ENS (top numerical model) on ChaosBench for deterministic, probabilistic, and physics-based metrics beyond Day 10.
Weaknesses
1. Aggressive compression (100×) may lose fine-scale features critical for regional forecasts (e.g., extreme events). No analysis of reconstruction error per variable.
2. Performance and efficiency actually depend a lot on the VAE’s reconstruction performance, compression ratio, and network structure. VAE has no statement in this regard, especially in experiments.
3. Trained on fixed 44-day sequences; performance degrades beyond this window (Appendix B.3). Unable to dynamically extend forecasts without retraining.
4. The scope of the comparative papers is too small. There are many excellent works now, but the authors did not compare.

---

> ### Author Rebuttal · Authors · 2025-07-30
>
> We thank the reviewer for the constructive feedback and for appreciating the design of SeasonCast and its strong performance in S2S prediction. We hope to address the reviewer’s questions and concerns below.
>
> > Aggressive compression (100×) may lose fine-scale features critical for regional forecasts (e.g., extreme events). No analysis of reconstruction error per variable.
>
> While we significantly compressed the spatial dimensions of the data to reduce the training sequence length for the transformer model, we actually increased the latent channel dimension. Specifically, each raw frame has a shape of 69x128x256, where 69 represents the number of weather variables. **Each latent frame has a shape of 1024x8x16, resulting in an effective compression ratio of (69x128x256) / (1024x8x16) = 17.25. Please refer to the tables below for the RMSE of the reconstruction across some key variables.**
>
> | T2m  | U10  | V10  | Z500  | T850 |
> |------|------|------|-------|------|
> | 0.71 | 0.43 | 0.40 | 27.34 | 0.57 |
>
> > Performance and efficiency actually depend a lot on the VAE’s reconstruction performance, compression ratio, and network structure. VAE has no statement in this regard, especially in experiments.
>
> We fully agree with the reviewer's observation that the VAE's performance is important to the overall efficacy of SeasonCast. Indeed, we dedicated substantial effort to developing a robust VAE model. Our primary goal was to achieve a high compression ratio along the spatial dimensions, as this directly reduces the number of training tokens required for the subsequent transformer model. Therefore, **we employed 16x spatial reduction across all experiments. We then incrementally increased the latent dimension until we obtained an acceptable reconstruction error.** Please refer to the table below for a detailed breakdown of performance across various latent dimensions.
>
> | D       | T2m  | U10  | V10  | Z500  | T850 |
> |---------|------|------|------|-------|------|
> | *D = 256*  | 0.96 | 0.65 | 0.62 | 48.72 | 0.77 |
> | *D = 512*  | 0.80 | 0.51 | 0.48 | 35.42 | 0.64 |
> | *D = 1024* | **0.71** | **0.43** | **0.40** | **27.34** | **0.57** |
>
> **Before selecting the specific VAE model presented in our paper, we also tried two alternative VAE architectures:**
> - VQ-VAE: This approach compresses data into a discrete latent space. However, we were unable to achieve satisfactory reconstruction errors; they were consistently 2 to 3 times higher than those obtained with a continuous VAE using the same spatial downsampling factor.
> - Video VAE: Similar to Cosmos [1], this variant compresses data across both spatial and temporal dimensions. Our early experiments showed that for an equivalent effective compression ratio, a per-frame VAE consistently outperformed a video VAE. This led us to opt for the per-frame VAE model.
>
> We thank the reviewer again for bringing this up and we will update the paper to discuss the VAE component in detail.
>
> [1] Agarwal, Niket, et al. "Cosmos world foundation model platform for physical ai." arXiv preprint arXiv:2501.03575 (2025).
>
> > Trained on fixed 44-day sequences; performance degrades beyond this window (Appendix B.3). Unable to dynamically extend forecasts without retraining.
>
> There appears to be a misunderstanding here. Figure 11b demonstrates that SeasonCast models trained on short windows exhibit performance degradation beyond their training window, whereas the SeasonCast model trained on the full 44-day window maintains consistent stability across all lead times. This finding is consistent with previous research indicating that autoregressive models typically struggle beyond the medium range. This also supports our assertion that training on the complete sequence of future frames is crucial for stabilizing the model in S2S prediction.
>
> **To further substantiate SeasonCast's ability to produce stable and well-calibrated predictions beyond its training window, we extended the model's rollout to 100 days ahead.** To achieve predictions beyond the initial training window, we iteratively employed the latest predicted frame (day 44) as the initial condition for predicting the subsequent window (day 45 to day 88), continuing this process until the desired lead time was reached. The table below presents SeasonCast's performance across key variables and various metrics at lead times exceeding 44 days. **SeasonCast remains exceptionally stable and well-calibrated, showing no signs of degradation even at day 100.**
>
> | Lead time (days) | RMSE (Z500) | RMSE (T850) | RMSE (Q700) | CRPS (Z500) | CRPS (T850) | CRPS (Q700) |
> |------------------|-------------|-------------|-------------|-------------|-------------|-------------|
> | 50               | 826.66      | 3.50        | 1.60        | 361.94      | 1.72        | 0.81        |
> | 70               | 821.64      | 3.46        | 1.60        | 358.96      | 1.70        | 0.81        |
> | 90               | 818.89      | 3.44        | 1.59        | 357.38      | 1.69        | 0.81        |
> | 100              | 812.41      | 3.44        | 1.59        | 355.44      | 1.69        | 0.81        |
>
> > The scope of the comparative papers is too small. There are many excellent works now, but the authors did not compare.
>
> Could the reviewer please elaborate on the specific baselines they have in mind? **We have made every effort to compare SeasonCast with state-of-the-art methods, including GC, PW, leading numerical models for the S2S setting, and GenCast and IFS-ENS for the medium-range settings**, and SeasonCast consistently demonstrates strong performance across benchmarks. We further compared it with Fuxi-S2S, another prominent method in S2S prediction. Please refer to our response to Reviewer mH3R for details on this experiment.
>
> > How does SeasonCast enforce physical laws (e.g., mass conservation) in generated forecasts? Are there constraints in the loss function or latent space?
>
> SeasonCast employs a purely learning-based approach, without imposing any physical constraints in the loss function for either the VAE or the transformer model. Incorporating physical laws is an orthogonal and promising direction that we aim to explore in future work.
>
> > Can SeasonCast predict rare events (e.g., heatwaves)? Please provide case studies on ChaosBench extreme indices.
>
> ChaosBench does not provide extreme case studies or indices, but instead provides a comprehensive evaluation framework across a wide range of metrics. SeasonCast consistently demonstrated strong performance within this benchmark, excelling across key variables, diverse metrics, and various lead times. Furthermore, its robust performance in the medium-range setting further highlights SeasonCast's versatile capabilities.

---

> > ### Author Response · Authors · 2025-08-04
> > **Reminder of our rebuttal**
> >
> > Dear Reviewer SUwF,
> >
> > We are writing to follow up on our rebuttal, in which we have done our best to address your concerns with additional experiments and detailed clarifications. As the discussion period closes in two days, we sincerely hope you have a moment to review our response. We are eager to address any remaining questions you might have and would be happy to discuss further. Thank you again for your valuable feedback.
> >
> > Best regards,
> >
> > The Authors

---

> > ### Comment · Reviewer_SUwF · 2025-08-07
> > **reply to author's rebuttal**
> >
> > Thank you for the author's rebuttal, which solved my questions. I will increase my review score.

---

> > > ### Author Response · Authors · 2025-08-07
> > > **Thank you for your review**
> > >
> > > We're glad to hear that our rebuttal has addressed your concerns. We truly appreciate your constructive feedback and for taking the time to provide a thoughtful review.
> > >
> > > Thank you for being willing to increase your score. Your suggestions regarding the VAE's reconstruction performance and the discussion on flexible forecasting windows are invaluable. We will incorporate these technical discussions and additional experimental results into the final version of our paper to improve its clarity and comprehensiveness.

---

> ### Author Response · Authors · 2025-08-06
> **Reminder of our rebuttal**
>
> Dear Reviewer SUwF,
>
> We are writing to follow up on our rebuttal, in which we have done our best to address your concerns with additional experiments and detailed clarifications. As the discussion period is coming to an end soon, we sincerely hope you have a moment to review our response. We are eager to address any remaining questions you might have and would be happy to discuss further. Thank you again for your valuable feedback.
>
> Best regards,
>
> The Authors

---

### Official Review · Reviewer_mH3R · 2025-07-10

**Clarity:** 2
**Significance:** 3
**Originality:** 2
**Rating:** 3
**Confidence:** 4

**Summary:**

Autoregressive architectures like GraphCast and PanguWeather struggle with subseasonal-to-seasonal (S2S) forecasting due to error accumulation across long horizons. This paper circumvents that bottleneck by replacing iterative rollouts with masked latent diffusion. A continuous-latent VAE compresses 69-channel ERA5 inputs into dense tokens. A bidirectional masked transformer, paired with a lightweight diffusion head, samples all masked space-time tokens in parallel, generating 44-day forecasts in a single pass. On ChaosBench, SeasonCast matches or exceeds ECMWF ensemble skill beyond day 15, while reducing inference cost by 10–20×. Ablations on masking strategy, latent dimension, temperature, and auxiliary loss suggest the method is robust under architectural perturbations.

**Questions:**

- Regarding the loss function:

    (a) How do you ensure the auxiliary MSE term does not bias the diffusion posterior, causing the ensemble mean to drift from the true predictive mean?

    (b) Do you need to adjust the relative weights between the two loss terms, if so, how? by calibration, ablation, or trial-and-error?

- FuXi-S2S generates daily means. Could you average SeasonCast outputs and evaluate against FuXi-S2S to give a rough comparison?
- For GenCast and NeuralGCM, could you run comparisons on scaled-down domains or resolutions to offer partial but informative benchmarks?

**Ethical Concerns:**

["NO or VERY MINOR ethics concerns only"]

**Limitations:**

The limitations of the methods are barely discussed.

**Quality:**

2

**Strengths And Weaknesses:**

he paper introduces a sound, practical approach, but the overall presentation falls short, and the technical novelty is modest. The main components (continuous VAE + masked-token transformer + diffusions) are standard in the literature. Their application to S2S is thoughtful, yet not fundamentally new. More importantly, the experimental analysis lacks depth. Metrics are plotted, but insight is scarce: the narrative rarely moves past curve comparisons, and the impact of architectural choices is not dissected. Without stronger analysis, the empirical story feels shallow. The paper shows promise, but in its current form, it is not ready for publication.

Strengths

- Sensible architectural design that addresses a real limitation in autoregressive systems.
- Demonstrated significant performance gains, particularly in the long-range regime.
- Efficient inference makes the method attractive for operational use.

 Weaknesses

**Clarity & Presentation**

- The manuscript is verbose in parts and under-explained in others.
- Background and methodology blur together; the structure lacks discipline.
- Notations appear before being defined (e.g., pUpU above Eq. (4)).
- Figure 1 omits key components and does not reflect the sampling pipeline; details are scattered rather than integrated.

**Technical Contribution**

- The core method, VAE + masked generative transformer, is standard in adjacent fields. The novelty lies in adaptation, not invention.
- The loss function blends diffusion loss with exponentially-weighted MSE for the mean prediction. This coupling feels ad hoc and risks undermining the benefits of probabilistic output that diffusion model provides.

**Empirical Analysis**

- The evaluation section reads more like a figure dump than an analysis.
- No statistical testing, no confidence intervals, no analysis of forecast reliability.
- The source of SeasonCast’s advantage over GraphCast or PanguWeather is left unexplored. If autoregressive error is to blame, it should be demonstrated explicitly in a controlled setting.

---

> ### Author Rebuttal · Authors · 2025-07-30
>
> We thank the reviewer for the constructive feedback, and for recognizing the significant performance and efficiency gains of SeasonCast. We hope to address the reviewer’s remaining concerns and questions below.
>
> > The manuscript is verbose in parts and under-explained in others. Background and methodology blur together; the structure lacks discipline.
>
> Since our methodology is built on top of existing techniques, i.e., masked generative modeling and diffusion models, we presented the background of these techniques in the Background section, and how we adapted them to SeasonCast in the Methodology section. **Can the reviewer point out which specific part of the manuscript is verbose and which is under-explained?** We’re happy to adjust the writing to make the paper clearer.
>
> > Notations appear before being defined (e.g., pUpU above Eq. (4)).
>
> $p_{\mathcal{U}}$ is the distribution of the binary mask, which we explain in more detail in Section 4.3 (lines 221-222). We will edit the paper to explain this clearly.
>
> > Figure 1 omits key components and does not reflect the sampling pipeline; details are scattered rather than integrated.
>
> We believe Figure 1 includes all key components of SeasonCast: the transformer backbone, the masking mechanism, the diffusion head, and the auxiliary deterministic head. We will further include an inference diagram in the next version of the manuscript.
>
> > The core method, VAE + masked generative transformer, is standard in adjacent fields.
>
> While our work adapts known techniques from other fields, we firmly believe that a well-executed adaptation can yield significant impact when applied to a new domain such as weather and climate. **We dedicated substantial effort to the design of SeasonCast to achieve state-of-the-art performance**, specifically through the full-sequence training and the integration of the proposed auxiliary loss in the masked generative model training. Furthermore, **the most impactful works in this field are themselves adaptation-focused**, including MetNet (which adapted a Video GAN model), GraphCast (which adapted GNN), and GenCast (which adapted GNN + a diffusion model).
>
> > The loss function blends diffusion loss with exponentially-weighted MSE for the mean prediction. This coupling feels ad hoc and risks undermining the benefits of probabilistic output that diffusion model provides.
>
> We respectfully disagree with the reviewer's assessment. **The deterministic and diffusion components operate as separate prediction heads atop the transformer backbone, thus preventing direct interference. Instead, the deterministic loss actively aids the training of the diffusion head**. This is because the deterministic loss provides an easy-to-optimize signal during training, enabling the model to learn meaningful $z_i$ that the diffusion head can effectively utilize for the probabilistic loss. Our ablation experiment in Figure 11a empirically demonstrates the positive impact of incorporating the deterministic loss across various metrics. Furthermore, our main experiments consistently show that SeasonCast remains well-calibrated across diverse scenarios, which further supports our argument.
>
> > The evaluation section reads more like a figure dump than an analysis. No statistical testing, no confidence intervals, no analysis of forecast reliability.
>
> Due to strict page limits, we focused our main comparisons on the most crucial baselines in the experiment section, which effectively highlighted SeasonCast's advantages. **For a more comprehensive understanding of SeasonCast's design, we included detailed ablation studies in Appendix B.3**. These studies thoroughly investigate the impact of various design choices, such as the deterministic loss, training sequence length, unmasking strategies, and diffusion sampling temperature.
>
> **Regarding statistical testing and confidence intervals, these are not standard practices in the field.** Training weather models on ERA5 is a computationally intensive process, especially given academic compute constraints. Leading papers like GraphCast, PanguWeather, Stormer, and GenCast have also omitted statistical testing and confidence intervals, making it unfair to penalize our paper on this basis.
>
> > The source of SeasonCast’s advantage over GraphCast or PanguWeather is left unexplored. If autoregressive error is to blame, it should be demonstrated explicitly in a controlled setting.
>
> Figure 4 clearly illustrates SeasonCast's superior performance beyond 10-day lead times, where GC and PW degrade due to autoregressive error accumulation. SeasonCast, however, maintains stability up to day 44. Further evidence from Figure 11b demonstrates that training SeasonCast with shorter future sequence lengths leads to increased error accumulation and poorer long-term performance. These experiments definitively show that SeasonCast's advantage in the S2S setting stems from its non-autoregressive nature and its ability to learn from extended sequences of future weather states, unlike other methods' one-step prediction approaches.
>
> > How do you ensure the auxiliary MSE term does not bias the diffusion posterior, causing the ensemble mean to drift from the true predictive mean?
>
> As we discussed above, the experiments in Figure 11a showed that the deterministic loss helps improve the performance of SeasonCast significantly across different metrics and lead times.
>
> > Do you need to adjust the relative weights between the two loss terms, if so, how? by calibration, ablation, or trial-and-error?
>
> We simply used equal weighting between the two terms without any tuning because we found that to work sufficiently well.
>
> >  Could you average SeasonCast outputs and evaluate against FuXi-S2S to give a rough comparison?
>
> We compared the daily forecasts from SeasonCast with Fuxi-S2S. As shown in the table below, SeasonCast demonstrates strong performance for 2-meter temperature and geopotential at 500hPa across four different lead times. Notably, Figure 8 in the Fuxi-S2S paper (https://arxiv.org/pdf/2312.09926) illustrates that the RMSE of Fuxi-S2S exceeds 2.45 for T2M and 800 for Z500 at lead times beyond 15 days. While their SSR scores are comparable, these results clearly indicate that SeasonCast either matches or surpasses Fuxi-S2S in both RMSE and SSR metrics. This further underscores the significant capabilities of SeasonCast, as it **outperforms Fuxi-S2S even without being specifically trained for daily average prediction, unlike Fuxi-S2S.**
>
> | Lead time (days) | RMSE (T2M) | RMSE (Z500) | SSR (T2M) | SSR (Z500) |
> |------------------|------------|-------------|-----------|------------|
> | 15               | 2.40       | 787.35      | 0.94      | 0.92       |
> | 25               | 2.40       | 794.64      | 0.93      | 0.91       |
> | 35               | 2.38       | 791.86      | 0.93      | 0.91       |
> | 40               | 2.37       | 794.81      | 0.94      | 0.91       |
>
> > For GenCast and NeuralGCM, could you run comparisons on scaled-down domains or resolutions to offer partial but informative benchmarks?
>
> As Figure 8 illustrates, both GenCast and NeuralGCM demand substantial computational resources. Even at a reduced resolution, generating a single 15-day forecast consumes several minutes. To evaluate these models on ChaosBench, we would need to run them for over a thousand initial conditions in the test set, producing 50 ensemble predictions for each, with a 44-day lead time. This extensive experiment is simply not feasible within the limited rebuttal period and the immense computational power it would necessitate.

---

> > ### Author Response · Authors · 2025-08-04
> > **Reminder of our rebuttal**
> >
> > Dear Reviewer mH3R,
> >
> > We are writing to follow up on our rebuttal, in which we have done our best to address your concerns with additional experiments and detailed clarifications. As the discussion period closes in two days, we sincerely hope you have a moment to review our response. We are eager to address any remaining questions you might have and would be happy to discuss further. Thank you again for your valuable feedback.
> >
> > Best regards,
> >
> > The Authors

---

> > > ### Comment · Reviewer_mH3R · 2025-08-06
> > >
> > > I’d like to thank the authors for the detailed response. My concerns, though, remain.
> > >
> > > Re: writing. As a concrete example, a sizable chunk of the introduction is dedicated to the description of the method, and similar content is repeated in the background section, which is supposed to be providing “background.”
> > >
> > > Re: loss function. My comments come from the calibration of the probabilistic forecast. Diffusion models naturally give you a probabilistic distribution, whose mean would likely differ from the deterministic (mean) header from the other component. How do you resolve such discrepancy while making sure the resulting distribution is calibrated?
> > >
> > > Re: empirical analysis. Deeper analysis is required to explain the results. As a concrete example, in the statement “clearly illustrates SeasonCast's superior performance beyond 10-day lead times, where GC and PW degrade due to autoregressive error accumulation”: why is it due to “autoregressive error accumulation”? A compelling experimental design would be something like: if you switch the predictive head of SeasonCast to be autoregressive, the performance deteriorates significantly. Such lack of depth in the “empirical study” is consistent across all experimental results.

---

> > > > ### Author Response · Authors · 2025-08-06
> > > > **Clarifications of important misunderstandings**
> > > >
> > > > We thank the reviewer for their continued engagement. We believe there may be a misunderstanding of key aspects of our work, which we would like to clarify.
> > > > - Regarding the Loss Function and Calibration: We want to respectfully clarify how our model architecture prevents the discrepancy you mentioned. **The deterministic head is a training-only auxiliary component. Its sole purpose is to provide an additional gradient that helps learn a meaningful latent representation $z_i$. This head is entirely discarded during inference, where only the diffusion head is used to generate the probabilistic forecast.** Therefore, there is no conflict between the two different heads. **Our main results and the ablation studies in Appendix B.2 demonstrate that this training strategy leads to excellent calibration** (comparable to the operational ECMWF-ENS system) and that the auxiliary deterministic loss is crucial for achieving state-of-the-art accuracy and calibration.
> > > > - Regarding the Depth of Empirical Analysis: We appreciate you suggesting a concrete experiment to validate our claim on autoregressive error accumulation. **We wish to clarify that we performed this exact analysis in Appendix B.3, Figure 11b.** In this ablation, we explicitly compared two different variants of the SeasonCast architecture: one that predicts the full sequence of 44 frames ahead, and the other that performs autoregressive predictions. The results clearly show that while the autoregressive models are competitive at short lead times, their performance degrades sharply after day 15, which directly supports our conclusion that our model's superiority at S2S scales stems from mitigating error accumulation. We believe this is a direct example of the in-depth analysis you are looking for.
> > > > - Regarding the Writing: Thank you for the feedback. We intended to use the Introduction to provide a high-level overview of the methodology for broader accessibility, while the Background section provides the formal, technical details and notations necessary for a rigorous understanding. We acknowledge the potential for repetition and will revise both sections in the final version to improve flow and clarity, ensuring the distinction between the two is sharper.

---

> > > > > ### Comment · Reviewer_mH3R · 2025-08-06
> > > > >
> > > > > Thanks for pointing out the ablation study in the appendix. It was my miss, apologize! As it's important to support your claim on the draw back of the autoregressive regime, I'd recommend moving it to the main text.
> > > > >
> > > > > Speaking of 11(c), I find it surprising that curves of auto regress and random are indistinguishable, do you have any hunch why? Taken the results as it is, it again raises the question whether "autoregressive regime" is the culprit. For the SSR plot, shouldn't it match the blue curve in Figure 6 (T850 (K))?

---

> ### Author Response · Authors · 2025-08-06
> **Clarifications of Figure 11**
>
> We sincerely thank you for the responsive feedback. We are glad our previous response was clarifying, and we will make sure to move the ablation study on the autoregressive regime into the main paper to improve visibility, as you recommended.
>
> Regarding your follow-up questions on Figure 11, we believe we can offer a clear explanation.
> - On the similarity of _Autoreg_ and _Random_ curves in Figure 11(c): The key is to distinguish the experiment in 11(c) from the one in 11(b). **The experiment in Figure 11(c) analyzes the impact of different unmasking strategies during inference on the same trained model, specifically, the 44f-24hr model that was trained to forecast the full sequence.** Because all three sampling strategies in 11(c) use the same full-sequence model, their overall performance is similarly strong. However, **the _Random_ curve is consistently better because using a different random unmasking order for each ensemble member introduces more forecast diversity**, which improves probabilistic scores like CRPS and SSR. We discussed this point in Lines 584-591 of the Appendix.
> - On the SSR plot in Figure 11 vs. Figure 6: Your observation of the discrepancy is correct, and it is intentional. **The difference is due to the sampling temperature hyperparameter $\rho $.** For all ablation studies in Figure 11, we used a default temperature of $\rho=1.0$ for a controlled analysis. For the main experiment in Figure 6, we used a temperature of $\rho=1.3$, which was tuned to optimize the overall performance on validation data.
> - This explains why the SSR in Figure 6 is higher (i.e., has more spread) than the SSR for the Random curve in Figure 11(c). This also provides a consistency check, as **the _MSE-10_ curve in Figure 11(a), _44f-24hr_ in 11(b), _Random_ in 11(c), and $\rho=1.0$ in 11(d) are all identical**, as they represent the same model with the same default inference settings.
>
> We hope this explanation fully clarifies these details. Thank you again for your diligence and for helping us strengthen the paper.

---

> > ### Author Response · Authors · 2025-08-07
> > **Follow up**
> >
> > Dear Reviewer mH3R,
> >
> > We are writing to follow up on our rebuttal. Have our clarifications regarding Figure 11 addressed your remaining questions? Since the discussion period is ending soon, we sincerely hope you will have time to respond to us and consider raising your rating of the paper. We are eager to address any remaining questions you might have and would be happy to discuss further. Thank you again for your valuable feedback.
> >
> > Best regards,
> >
> > The Authors

---

> > > ### Comment · Reviewer_mH3R · 2025-08-07
> > >
> > > Thanks for the response. I'm still not convinced that the empirical evidence supports the claim that the autoregressive component is the one to blame. My broader concerns about the depth of the empirical analysis, technical contributions, and presentation remain. I'll keep my rating. Still, I'd like to thank the authors again for the discussions.

---

> > > > ### Author Response · Authors · 2025-08-07
> > > >
> > > > We thank the reviewer for the engaging discussion and for sharing their final assessment. While we respect your position, we believe there is a significant disconnect between your remaining concerns and the evidence presented in our paper and rebuttal. For the record, we would like to offer this final summary to clarify why our claims are strongly supported.
> > > > - On the Autoregressive Claim: **We are confident the empirical evidence is conclusive, primarily from the ablation study you asked us to perform.** The experiment isolates the variable perfectly: we compare two versions of SeasonCast with the exact same architecture, where the only difference is the training horizon (full-sequence vs. autoregressive). The result clearly showed that the autoregressive model accumulates errors and collapses after day 15, matching the behavior of PanguWeather and GraphCast. In contrast, the full-sequence model's skill remains stable. This provides direct causal evidence that the autoregressive regime is the source of failure at S2S scales. **This was also well-documented in the literature.** As quoted in ChaosBench, "we find models trained directly (e.g., ViT/ClimaX) have better performance than those used autoregressively (e.g., PW, GC, FCN2). This suggests that error propagation is a significant source of error, and controlling for stability is key to extend the predictability range".
> > > > - On Technical Contributions: We respectfully assert that our contributions are significant. We propose a novel method that is **one of the first to achieve state-of-the-art performance at S2S prediction**, matching the operational ECMWF-ENS system across accuracy, physical consistency, and probabilistic calibration. **Furthermore, SeasonCast remains competitive with the IFS-ENS system and GenCast at medium-range while being $10\times $ to $20\times $ faster.** We believe these results represent a substantial advance for the field.
> > > > - On the Depth of Empirical Analysis: We respectfully disagree with the idea that our analysis lacks depth. **We performed extensive ablation studies to investigate every crucial component of SeasonCast:** the impact of the deterministic objective, the training sequence length, different unmasking strategies, sampling temperature, the number of ensemble members, and the impact of noise in the initial conditions. We believe this is a clear demonstration of a comprehensive and deep empirical study.
> > > > - On Presentation: We acknowledge your feedback on the writing. **We view this as a minor organizational issue, as the paper already contains all the necessary information to understand the method's motivation and technical details.** It would be a straightforward task to incorporate your suggestions to rearrange sections for improved clarity in the final version.

---

### Note · Authors · 2025-08-15

We sincerely thank the reviewers, AC, and SAC for their time and constructive discussions. We are encouraged that reviewers recognized SeasonCast's novel and sensible design (pne4, Er6Q), significant performance and efficiency gains in S2S and medium-range forecasting (mH3R, SUwF), and the strength of our results (pne4).

The discussions focused on several key points, which we believe we have fully addressed:
- Depth of Empirical Analysis: The most critical discussion point was whether our claim about mitigating autoregressive error was sufficiently supported. We clarified that a direct, controlled ablation study isolating this effect was indeed present in Appendix B.3 (Fig. 11b), a point that was initially missed. We have committed to moving this crucial experiment to the main text to improve its visibility.
- Technical Soundness: Concerns were raised about our auxiliary loss potentially biasing the probabilistic output. We clarified a key aspect of our design: the deterministic loss has a separate prediction head and is a training-only component, entirely discarded during inference. This design prevents any conflict between the two heads and, as our ablations show, is crucial for achieving our state-of-the-art accuracy and calibration.
- Completeness of Evaluation: In response to feedback (pne4, Er6Q, SUwF), we provided new results during the rebuttal, including: detailed VAE ablation studies, ACC metrics for key variables, and performance on important surface-level variables. We have also committed to adding more qualitative results to the final manuscript.
- S2S Performance Context: We addressed concerns about performance relative to climatology (Er6Q) by contextualizing that while deterministic skill is theoretically limited beyond 14 days, SeasonCast's primary value is providing robust, well-calibrated probabilistic forecasts where climatology fails to offer.

Overall, we believe the review process has been highly productive. We have clarified key misunderstandings and addressed all major technical and empirical concerns through our rebuttals. We are confident that SeasonCast represents a significant advance for S2S forecasting and that the final paper will be substantially stronger thanks to this feedback.

---

### Decision · Program_Chairs · 2025-09-17

**Decision:**

Accept (poster)

**Comment:**

This paper explores the use of a masked transformer in the latent space of a variational autoencoder for weather prediction. A combination of a diffusion loss and an MSE loss is used to learn a distribution over masked tokens representing future states. The initial reviews of this paper were mixed, with an average rating of 3.75. On the positive side, the reviewers acknowledged the architectural design and promising performance in long-range forecasts. The diffusion loss and inherent circumvention of error accumulation through sequence prediction, in particular, were perceived as strengths of the proposed approach. Although the architecture's novelty is modest, it is appealing and well-executed from a design point of view. Supporting the diffusion loss with a short-range MSE loss was demonstrated to be beneficial. The feedback on the quality of the manuscript was mixed, with some reviewers expressing concerns about its structure and the lack of more qualitative insights beyond the quantitive evaluation. Performance-wise SeasonCast compares favourably to two deep learning approaches and competitive relative to more computationally expensive numerical models, especially in the long-term regime. In their rebuttal the authors addressed many of the raised concerns, proposing improvements to the manuscript and highlighting the benefits and contributions of a non-autoregressive approach. Additional experiments, such as a comparison to FuXi-S2S, fine-grained reconstruction errors, longer lead times up to 100 days, and a (preliminary) evaluation of the Anomaly Correlation Coefficient (ACC) provide additional context around SeasonCast’s performance.

Overall, SeasonsCast is a well-designed architecture with demonstrated strengths at its targeted forecasting horizon. The reported experiments show advantages over deep learning-based alternatives and validate the design choices in a series of insightful ablation studies. Additional comparisons to established numerical models show competitive performance at faster inference runtimes. The paper is therefore recommended for acceptance as a poster, with the expectation that the feedback received during the review process will be incorporated into the camera-ready manuscript. Including comparisons to Gencast/NeuralGCM and improving the coverage of the runtime analysis, in particular, would strengthen the paper.